# Poly(ADP-ribose) polymerase 1 accelerates vascular calcification by upregulating Runx2

Cheng Wang[1,2,3], Wenjing Xu[1,2], Jie An[1,2], Minglu Liang[1], Yiqing Li[4], Fengxiao Zhang[1,2], Qiangsong Tong[1] & Kai Huang[1,2]

Vascular calcification is highly prevalent in end-stage renal diseases and is predictive of cardiovascular events and mortality. Poly(ADP-ribose) polymerase 1 (PARP1) inhibition or deletion is vasoprotective in several disease models. Here we show that PARP activity is increased in radial artery samples from patients with chronic renal failure, in arteries from uraemic rats, and in calcified vascular smooth muscle cells (VSMCs) in vitro. PARP1 deficiency blocks, whereas PARP1 overexpression exacerbates, the transdifferentiation of VSMCs from a contractile to an osteogenic phenotype, the expression of mineralization-regulating proteins, and calcium deposition. PARP1 promotes Runx2 expression, and Runx2 deficiency offsets the pro-calcifying effects of PARP1. Activated PARP1 suppresses miRNA-204 expression via the IL-6/STAT3 pathway and thus relieves the repression of its target, Runx2, resulting in increased Runx2 protein. Together, these results suggest that PARP1 counteracts vascular calcification and that therapeutic agents that influence PARP1 activity may be of benefit to treat vascular calcification.

---

[1] Clinical Center for Human Genomic Research, Union Hospital, Tongji Medical College, Huazhong University of Science and Technology, Wuhan 430022, China. [2] Department of Cardiovascular Diseases, Union Hospital, Tongji Medical College, Huazhong University of Science and Technology, Wuhan 430022, China. [3] Department of Rheumatology, Union Hospital, Tongji Medical College, Huazhong University of Science and Technology, Wuhan 430022, China. [4] Department of Vascular Surgery, Union Hospital, Tongji Medical College, Huazhong University of Science and Technology, Wuhan 430022, China. These authors contributed equally: Cheng Wang, Wenjing Xu. Correspondence and requests for materials should be addressed to C.W. (email: cwangunion@hust.edu.cn) or to K.H. (email: huangkaiunion@gmail.com)

Vascular calcification is highly prevalent in chronic renal failure (CRF) and is associated with further cardiovascular morbidity and mortality[1–3]. Calcification rapidly progresses in patients on dialysis. Ectopic calcium deposition in the vasculature contributes to vessel wall stiffening and loss of elastic recoil, thus resulting in unstable hemodynamic consequences and finally leading to a decline in end-organ perfusion and subsequent ischaemic events and heart failure[1,4].

However, no ideal approaches exist to prevent or reverse vascular calcification, partly because the mechanisms are heterogeneous and complex. Previously, vascular calcification was thought to be a passive process involving the excess precipitation of calcium-phosphate minerals in vascular tissue. A more recently recognized characteristic of the disease is that arterial calcification is an active, complicated and regulated process[5,6]. Ectopic vascular calcification follows a process similar to physiologic bone formation. Vascular smooth muscle cells (VSMCs) are of mesenchymal origin and, under stress, can differentiate into different mesenchymal-derived cell types, such as osteoblasts and chondrocytes, leading to calcification, altered matrix production, and lipid accumulation. At sites of calcification, VSMCs undergo an osteochondrocytic phenotypic change and upregulate the expression of mineralization-regulating proteins, thus contributing to vascular calcification[7,8].

Poly(ADP-ribose)polymerase (PARP) 1 is a dominant member of the PARP family in eukaryotes, accounting for approximately 90% of cellular PARP activity. Upon activation, PARP1 attaches ADP-ribose polymer chains to target proteins using nicotinamide adenine dinucleotide (NAD$^+$) as a substrate and facilitates the process of DNA repair[9]. However, increasing evidence demonstrates that PARP1 can go beyond DNA repair and is involved in a wide range of cellular and biological processes[10], including chromatin compaction, stress signaling, cell death, inflammation, and differentiation[9,11–13]. Furthermore, defects in PARP1 function have been linked to diseases, such as chronic inflammation, neurodegenerative disorders, cardiovascular diseases, and cancer[14–16]. However, the role and mechanism of PARP1 in vascular calcification are poorly understood.

Oxidative stress has emerged as a constant feature of CRF[17,18]. Accumulating data indicate that oxidative stress is associated with dysfunction of various organs, including arterial medial calcification in chronic kidney disease (CKD). PARP1 functions as an oxidative stress sensor that propagates stress signals to execute downstream molecular actions[9], thus leading to multiple physiological and pathological processes. Although Edit Nagy et al. recently found an increased transcript level of PARP1 in the interstitial cells of human calcified tricuspid aortic valves[19,20], the valvular calcification is quite different from vascular calcification, including architecture and the resident cell populations. Therefore, the detailed contributions of PARP1 to vascular calcification and VSMC osteogenesis in CRF still remain unknown.

Runt-related transcription factor-2 (Runx2), a key regulator of osteoblast differentiation and bone development, plays an important role in vascular calcification and can induce transdifferentiation of VSMCs to an osteochondrocytic phenotype[21–23]. In normal vascular cells, Runx2 expression is nearly undetected but is dramatically elevated in calcified vascular tissue specimens from atherosclerotic plaques or uraemic arteries[24]. The expression and activity of Runx2 are regulated by several signaling pathways, such as TGFβ/BMP2 and Wnt[25,26]. Moreover, aberrant post-translational modifications of Runx2, such as microRNA (miRNA)-dependent control[27], acetylation or methylation, also play critical roles during vascular calcification.

In this study, our results show that manipulation of PARP1 expression can dramatically interfere with VSMC osteogenic transition and vascular calcification both in vivo and in vitro. We screen Runx2 as the downstream target, and discover that activated PARP1 promotes Runx2 expression at the post-transcriptional level via the IL-6/STAT3/miR-204 pathway, suggesting a key role for PARP1 in vascular calcification and new approaches for potential therapy.

## Results

**PARP activity is increased during vascular calcification.** Oxidative stress is enhanced in uraemia[17,28,29] (Supplementary Fig. 1) and PARP1 can be activated by multiple cellular stresses, especially ROS, so it is rational to speculate that PARP1 can be evoked during vascular calcification as a result of CRF. To confirm this hypothesis, PARP activity was tested in radial artery specimens from uraemic patients who underwent an arterial venous fistula operation[30]. Compared to normal controls, increased total poly(ADP-ribosyl)ation levels and PARP1 expression were observed in CRF specimens (Fig. 1a), and vascular PARP activity was also significantly increased (Fig. 1b), indicating that PARP was activated in the CRF arterial specimens. Next, we quantified PARP activity and the extent of arterial calcification. As shown in Fig. 1c, a positive correlation was observed between PARP activity and the extent of calcium content ($R^2 = 0.4463$, $P < 0.05$, Pearson's correlation coefficient analysis), suggesting that PARP activation is associated with the severity of arterial calcification. A rat CRF model was generated by feeding Wistar rats a 0.75% adenine diet for 6 weeks to mimic arterial medial calcification, and a significant increase was consistently observed in the plasma levels of blood urea nitrogen (BUN) and creatinine (Cr) (Supplementary Table 1). Along with increased aortic calcification and calcium deposition, as evidenced by the morphology (Supplementary Fig. 2a), von Kossa staining and calcium quantification (Fig. 1d and Supplementary Fig. 2b, 12a), respectively, immunofluorescence staining showed a clear upregulation of aortic poly(ADP-ribosyl)ation levels in CRF rats compared to chow-fed controls (Fig. 1d). To confirm this finding, mineralization of rat aortic rings was successfully induced in osteogenic media containing 10 mmol/L β-glycerophosphate (βGP) for different periods[31] (Supplementary Fig. 2c, d). High Pi conditions increased the calcium content and corresponding PARP activity in aortic rings in a time-dependent manner (Fig. 1e). These observations were verified by rat VSMC calcification in vitro, which showed similar trends as the calcium content, alkaline phosphatase (ALP) activity and corresponding PARP activity (Fig. 1f–h). Taken together, these data suggest the relevance of activated PARP in vascular calcification.

**PARP1 deficiency attenuates vascular calcification.** Because PARP1 is activated in calcified arteries, we hypothesized that inhibition of PARP1 activity might have beneficial effects on the onset and development of vascular calcification. Abdominal aortas were inoculated with adenovirus carrying PARP1 (PARP1 shRNA) or scrambled shRNA (Scr shRNA) after 3 weeks of the adenine diet. Six weeks after induction of the diet, specific knockdown of PARP1 in aortas was verified at the protein level (Fig. 2a), along with the serum biochemical parameters (Supplementary Table 2). Strikingly, PARP1 knockdown markedly decreased calcified nodule formation and calcium deposition compared to the scrambled group (Fig. 2b, c and Supplementary Fig. 12b), suggesting that deficiency of PARP1 alleviates CRF-induced vascular calcification. In support of this supposition, we found that treatment of CRF rats with the PARP inhibitor 3AB also attenuated aortic calcium deposition, as demonstrated by von Kossa staining and a calcium content assay (Supplementary Fig. 3a, b and Supplementary Fig. 12c). Because vascular calcification is mainly mediated by VSMCs, we further examined the

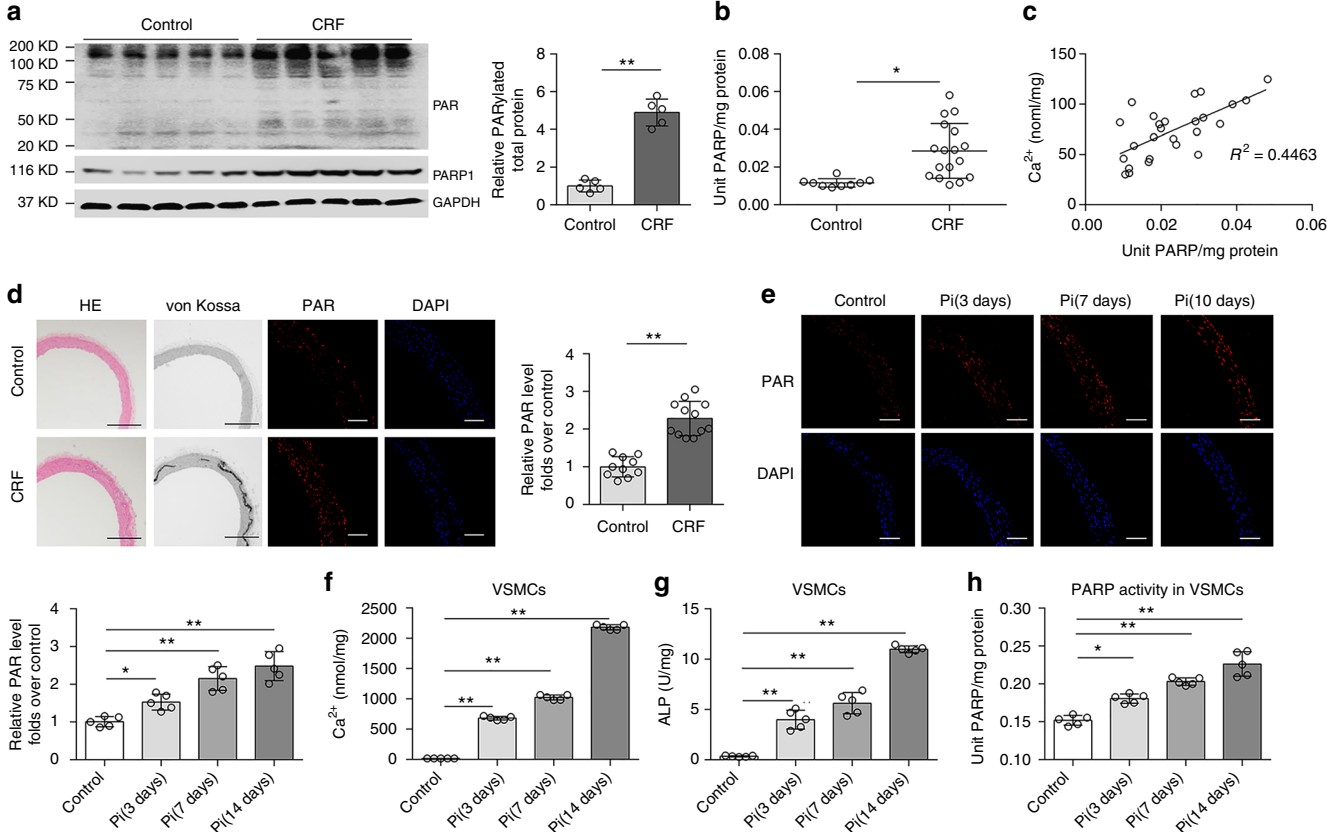

**Fig. 1** PARP activity is increased in calcifying arteries and VSMCs. **a** Representative western blot analysis of PARP1 expression and PARP activity (Poly (ADP-ribosyl)ation) in radial arteries. Chronic renal failure (CRF): patients with CRF and uremia who underwent arterial venous fistular surgery ($n = 5$); Control ($n = 5$): patients who underwent amputation surgery due to arm trauma, without diagnosed complications of diabetes and chronic kidney disease. **b** PARP activity in radial arteries of control ($n = 9$) and CRF patients ($n = 17$) were assayed using the universal colorimetric PARP assay kit. **c** The calcium content and PARP activity were positively correlated in CRF arteries. $R^2 = 0.4463$, $P < 0.05$. Statistical significance of correlations was determined by Pearson's correlation coefficient analysis. **d** Wistar rats were fed an adenine diet or a chow diet for 6 weeks ($n = 10$–12 per group). Arteries were isolated and performed by hematoxylin/eosin (H&E) and von Kossa staining. The PAR level was determined by immunofluorescence staining. Scale bars: black, 200 μm; white, 100 μm. **e** Rat aortic rings were treated with high Pi (osteogenic medium containing 10 mM β-glycerophosphate) for indicated days (0, 3, 7, and 10 days), and PARP activity was then detected by immunofluorescence staining. ($n = 5$ per group). Scale bar, 100 μm. **f**–**h** Rat VSMCs were treated with high Pi (10 mM β-glycerophosphate) for indicated days (0, 3, 7, and 14 days). The calcium content (**f**), ALP activity (**g**), and PARP activity (**h**) in calcified VSMCs were assayed. ($n = 5$ per group). Statistical significance was assessed using one-way ANOVA for multiple comparison and two-tailed $t$-tests for two groups and is presented as follows: *$P < 0.05$ and **$P < 0.01$. All values are means ± S.D.

effects of PARP1 on atrial calcification in cultured primary VSMCs induced by βGP. Knockdown of PARP1 or treatment with the PARP inhibitor 3AB significantly reduced the intensity of Alizarin red S staining, calcium deposition and the corresponding ALP activity (Fig. 2d–f and Supplementary Fig. 3c–e).

**PARP1 overexpression aggravates vascular calcification**. We next examined whether overexpression of PARP1 could promote VSMC calcification. We delivered Ad-PARP1 or Ad-Null to CRF rat aortas, and PARP1 overexpression was verified as indicated in Fig. 2g, along with the serum biochemical parameters (Supplementary Table 3). Ad-Null infection did not alter the calcification of the abdominal aorta. In contrast, Ad-PARP1 greatly aggravated aortic calcification, as evidenced by von Kossa staining and calcium quantification (Fig. 2h, i and Supplementary Fig. 12d). We then evaluated the effects of PARP1 overexpression on cultured VSMCs in vitro. When exposed to high βGP, Ad-PARP1 clearly exacerbated the calcium deposition in VSMCs compared to Ad-Null (Fig. 2j–l). Thus, these results suggest that PARP1 promotes vascular calcification.

**PARP1 promotes the osteogenic transition of VSMCs**. VSMC calcification is concomitant with osteochondrocytic phenotypic changes, which are characterized by the induction of osteo-chondrogenic transdifferentiation markers and suppression of smooth muscle cell lineage markers (Supplementary Fig. 4), and require the uptake of phosphate by sodium-dependent phosphate co-transporters (PIT1 and PIT2). We found that PARP1 deficiency had no effects on the expression of PIT1 or PIT2 (Fig. 3a, b), but surprisingly interfered with the osteogenic transition of VSMCs. Expression of the osteogenic markers osteocalcin (OC), collagen IA1 (ColIA1), and osteopontin (OPN) was markedly depressed in calcified VSMCs with PARP1 knockdown (Fig. 3c) or treated with PARP1 inhibitors (Fig. 3d), while the levels of the contractile markers of SMA, SM22 and SMMHC were concomitantly reversed. In contrast, forced expression of PARP1 resulted in increased osteogenic gene expression but suppressed contractile gene expression (Fig. 3e). Moreover, in human aortic SMCs (HASMCs), either PARP inhibitor or PARP1 deficiency could attenuate the βGP-induced osteogenic transition together with the decreased calcium deposition (Supplementary Fig. 5). Combined, these data show that activated PARP1 promotes the

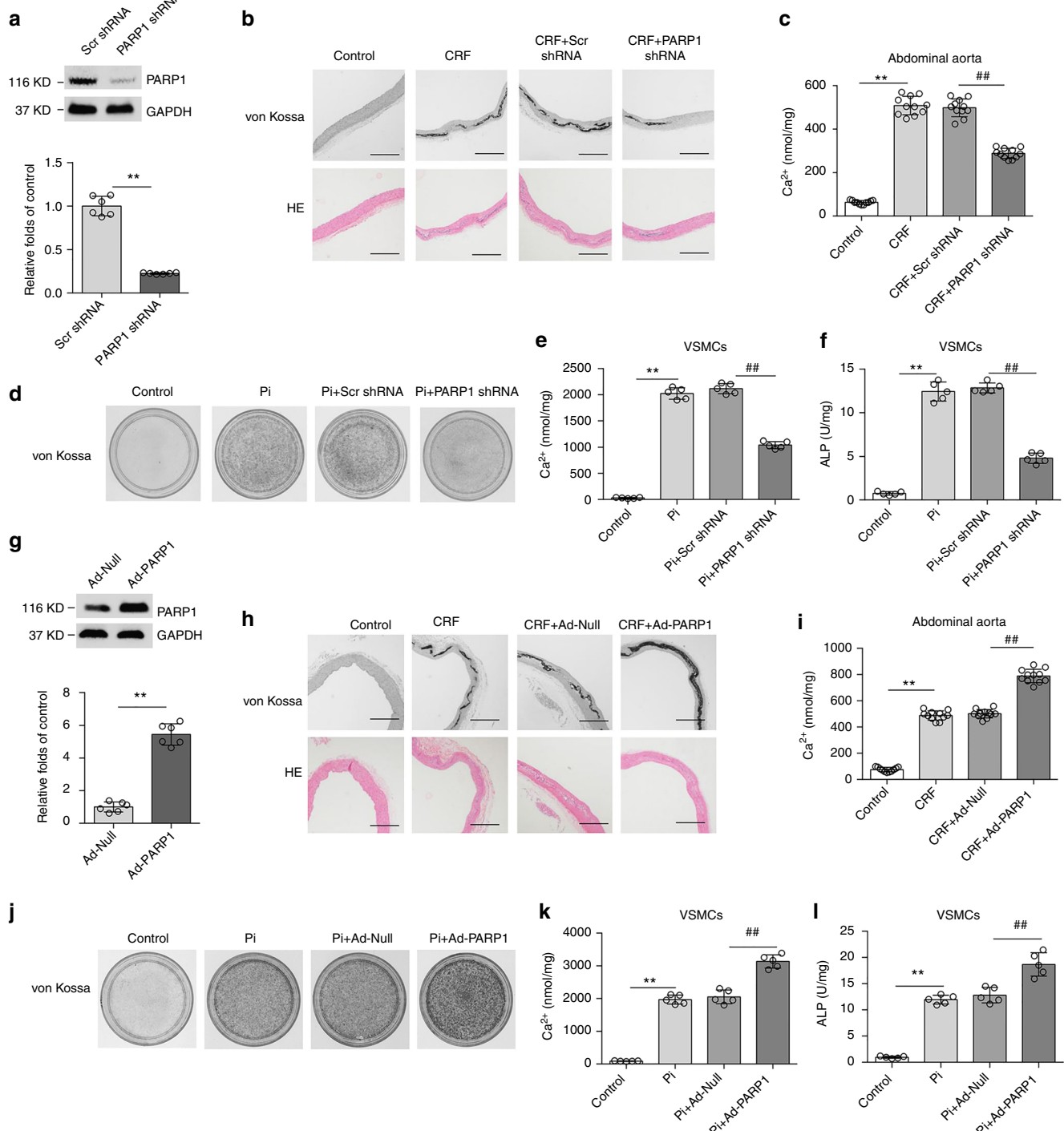

**Fig. 2** Manipulation of PARP1 expression regulates vascular calcification. **a–c** Rat abdominal aortas were inoculated with adenovirus encoding Scrambled (Scr shRNA) or PARP1 shRNA at three weeks after the adenine diet, and then fed for three weeks. PARP1 deficiency in arteries was identified by western blot (**a**) ($n = 5$ per group). Aortas were stained by H&E and von Kossa for mineralization (**b**), and the calcium deposition in arteries was quantified (**c**). ($n = 10$–12 per group). Scale bar, 100 μm. **d–f** Rat primary VSMCs were pre-infected with Scrambled or PARP1 shRNA adenovirus and then exposed to osteogenic medium for 14 days. VSMCs were stained for mineralization by Alizarin red S (**d**), and the quantitative analysis of calcium content (**e**) and ALP (**f**) were detected respectively. ($n = 5$ for each group). **g–i** Rat abdominal aortas were inoculated with Ad-Null or Ad-PARP1 at three weeks after the adenine diet, and then fed for three weeks. PARP1 overexpression in arteries was evaluated via western blot (**g**) ($n = 5$ per group). Aortas were stained by von Kossa and H&E for mineral nodules (**h**) and the quantification of calcium deposition was calculated (**i**). ($n = 10$–12 per group). Scale bar, 100 μm. **j–l** Rat primary VSMCs were pre-infected with Ad-Null or Ad-PARP1 adenovirus and then exposed to osteogenic medium for 14 days. Alizarin red S staining (**j**), calcium content (**k**), and ALP (**l**) in calcified VSMCs were then determined. ($n = 5$ per group). Statistical significance was assessed using one-way ANOVA for multiple comparison and two-tailed $t$-tests for two groups and is presented as follows: $**P < 0.01$ and $^{##}P < 0.01$. All values are means ± S.D.

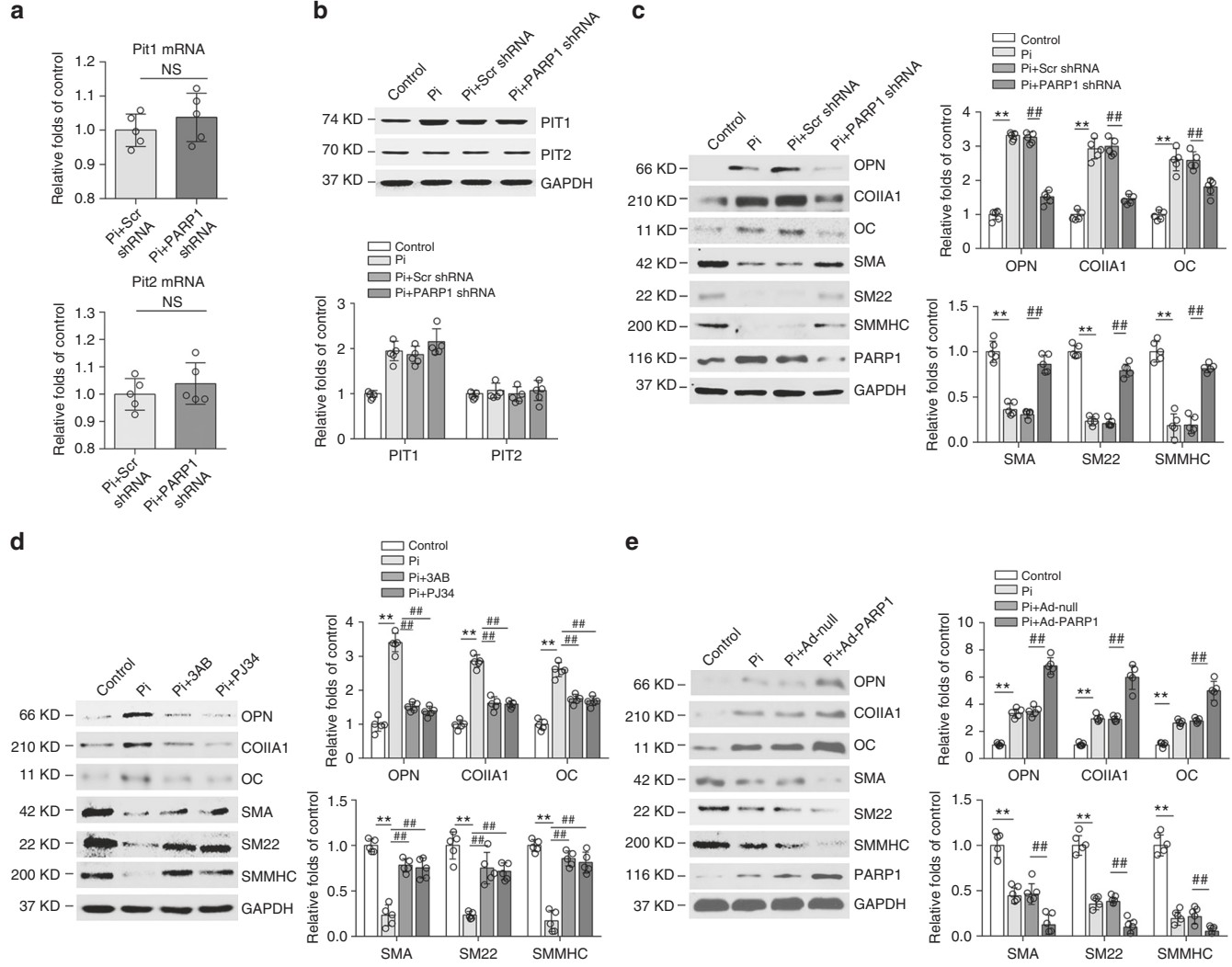

**Fig. 3** PARP1 regulates VSMCs osteogenic transition, not Na/P co-transporter. **a, b** Quantitative RT-PCR (**a**) and western blot (**b**) analysis of PIT1 and PIT2 expression in VSMCs, which were infected with Scrambled (Scr shRNA) or PARP1 shRNA, and treated with high Pi. **c–e** Rat VSMCs were infected with adenovirus carrying Scr shRNA, PARP1 shRNA, Null or PARP1 gene for 48 h and then incubated with osteogenic media for 14 days. PARP1 inhibitors 3AB (10 mM) and PJ34 (10 μM) were added every day. Western blot analysis and related quantification of osteogenic-related marker genes (OPN, ColIA1, and OC) and smooth muscle marker genes (SMA, SM22 and SMMHC) expression in VSMCs with PARP1 knockdown (**c**), PARP1 inhibitors 3AB and PJ34 (**d**), and PARP1 overexpression (**e**). ($n = 5$ for each group). Statistical significance was assessed using one-way ANOVA for multiple comparison and two-tailed $t$-tests for two groups and is presented as follows: NS: no significance, \*\*$P < 0.01$ and ^{##}$P < 0.01$. All values are means ± S.D.

osteogenic transition of VSMCs and subsequent vascular calcification.

**PARP1 aggravates vascular calcification via Runx2.** Osteogenic transdifferentiation of VSMCs relies on multistep molecular pathways regulated by different transcription factors and signaling proteins, such as Runx2, MSX2, SOX9, DLX5, and OSTERIX[3,32,33]. After screening, only Runx2 expression was subject to PARP1 regulation. Western blot and immunofluorescence staining analyses revealed that PARP1 knockdown suppressed the elevated expression of Runx2 in cultured VSMCs (Fig. 4a) and in the abdominal aortas of CRF rats (Fig. 4b), accompanied by reciprocal changes in SMA.

To explore whether Runx2 mediates the VSMC calcification promoted by PARP1, rat abdominal aortas were transfected with adenovirus carrying Runx2 shRNA and Ad-PARP1 (Supplementary Table 4). Although PARP1 markedly enhanced vascular calcification, Runx2 deficiency effectively suppressed the vascular

calcification, as evidenced by the von Kossa staining results and calcium content, and the related osteogenic marker expression (Fig. 4c–e and Supplementary Fig. 12e). Consistent with this observation, Runx2 depletion also dramatically abrogated the increase in calcium deposition by PARP1 in cultured artery rings and VSMCs (Fig. 4f, g), indicating that Runx2 may be essential for PARP1-induced vascular calcification.

Considering that PARP1 was not activated without phosphate treatment (Supplementary Fig. 6a), we chose the PARP1 L713F mutant, which is constitutively active in cells[34–36], to assess the dependence of PARP1 on Runx2 without phosphate treatment. Interestingly, WT PARP1 could not directly induce osteogenic gene expression or calcium deposition, but L713F mutant could induce these changes. When Runx2 was depleted, the above effects of PARP1 L713F were suppressed (Supplementary Fig. 6b, c). Moreover, we further overexpressed Runx2 in the presence of PARP1 deficiency in vitro. As shown in Supplementary Fig. 7, ectopic Runx2 antagonized the protective effects of PARP1

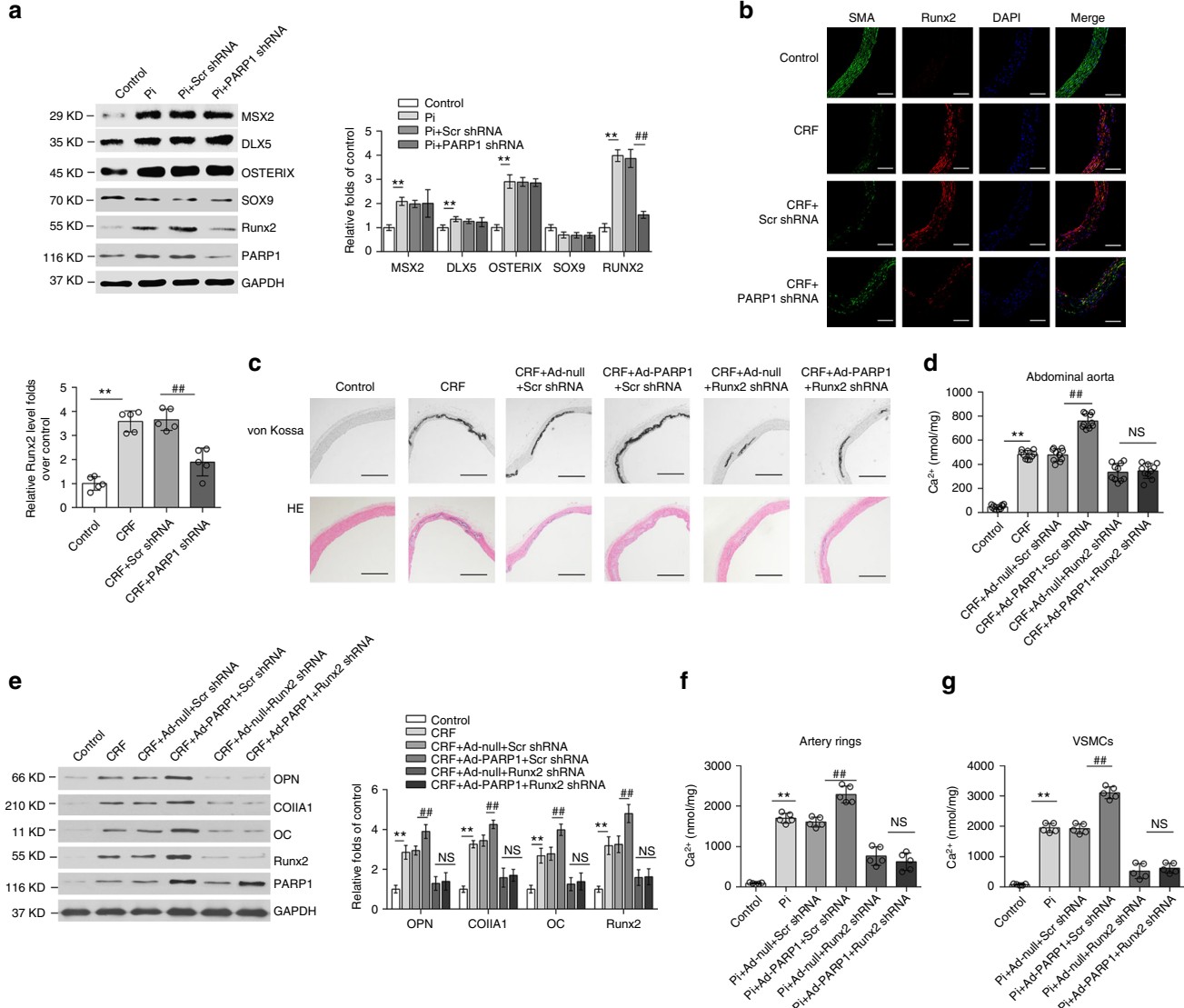

**Fig. 4** Runx2 acts as a mediator of the pro-calcifying effects of PARP1. **a** Western blot analysis of several osteogenic factors expression in calcified VSMCs transfected with Scr or PARP1 shRNA. ($n = 5$ per group). **b** The expression levels of Runx2 and SMA in abdominal arteries of indicated groups were determined by immunofluorescence staining. ($n = 5$ per group). Scale bar, 100 μm. **c–e** Rat abdominal aortas were inoculated with Ad-Scr shRNA, or Ad-Runx2 shRNA together with Ad-Null or Ad-PARP1 at three weeks after the adenine diet. Six weeks later, arteries were isolated and the calcification was analyzed by H&E, von Kossa staining (**c**) and the calcium quantification (**d**) ($n = 10–12$ per group), and the downstream osteogenic markers (OPN, ColIA1, OC, and Runx2) were analyzed by western blot (**e**). ($n = 5$ per group). Scale bar, 100 μm. **f, g** The aortic rings (**f**) or rVSMCs (**g**) were pre-infected with Ad-Scr shRNA, or Ad-Runx2 shRNA together with Ad-Null or Ad-PARP1, and exposed to the osteogenic medium for 14 days. The calcification was determined by the calcium assay respectively. ($n = 5$ for each group). Statistical significance was assessed using one-way ANOVA for multiple comparison and is presented as follows: NS: no significance, **$P < 0.01$ and ##$P < 0.01$. All values are means ± S.D. Source data are provided as a Source Data file

deficiency on Pi-induced VSMC calcification. These data suggest a direct dependence of PARP1 on Runx2 for vascular calcification.

**PARP1 enhances Runx2 expression via suppression of miR-204.** We further explored the specific mechanisms underlying the regulation of Runx2 by PARP1. As PARP1 is involved in the transcriptional regulation of multiple genes, we first sought to evaluate whether PARP1 could influence the transcriptional activity of the Runx2 promoter. We transfected A7r5 cells with the Runx2 promoter (pGL3-Runx2-luc) and subjected cells to the indicated treatments. The luciferase assay results showed that neither PARP1 knockdown nor overexpression altered Runx2 promoter activity (Fig. 5a).

miRNAs (or miRs), targeting the mRNA 3′ untranslated regions (3′UTR) for translational repression or mRNA degradation[37], have emerged as an alternative method of gene regulation. We fused the full-length Runx2 3′UTR to the luciferase reporter (pMIR-REPORT) and transfected it into A7r5 cells. Surprisingly, high Pi conditions dramatically increased Runx2 3′UTR luciferase activity, while PARP1 inhibition or knockdown comparably suppressed the increased activity (Fig. 5b). We also performed quantitative reverse transcription-polymerase chain reaction (qRT-PCR) in cultured VSMCs and found no clear changes in *Runx2* mRNA with PARP1 manipulation (Supplementary Fig. 8), suggesting that miRs might participate in the regulation of Runx2 via PARP1 through translational repression.

Next, we performed bioinformatics analyses using the TargetScan, miRanda, and miRwalk programmes to search for

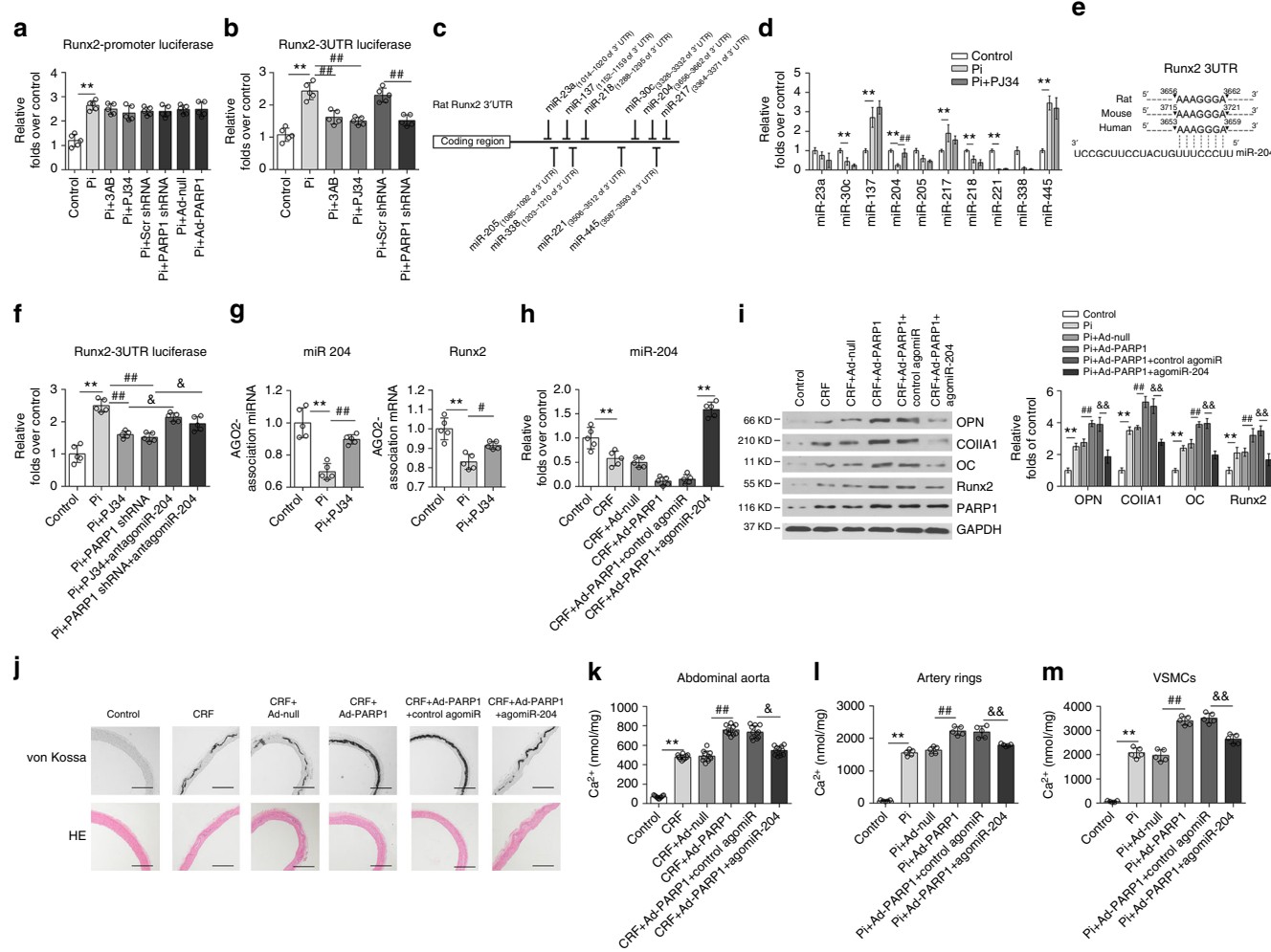

**Fig. 5** miR-204 mediates the regulation of PARP1 on Runx2. **a** A7r5 cells were transfected with the Runx2 promoter, along with 3AB, PJ34 or Scr shRNA and PARP1 shRNA treatments. The luciferase activity was analyzed. (n = 5 per group). **b** The Runx2-3′UTR plasmid was transiently transfected into A7r5 cells before stimulation as indicated. The luciferase reporter assay was analyzed. (n = 5 per group). **c** Seed regions of predicted Runx2-targeting miRNAs. **d** Selected miRNA expression levels in calcified VSMCs were analyzed by qRT-PCR assays. (n = 5 per group). **e** The putative miR-204 sequence present in Runx2 is conserved across different species. **f** A7r5 cells were co-transfected with the Runx2-3′UTR plasmid and miR-204 antagomir, and then treated with high Pi in the absence or presence of PARP inhibitor or PARP1 deficiency. The luciferase activity was analyzed. (n = 5 per group). **g** Rat VSMCs were treated with high Pi or high Pi + PJ34 for 3 days. The binding ability of miR-204 to Runx2 3′UTR in RISC complex was detected by RIP assay. **h–k** Rat abdominal arteries were inoculated with Ad-Null or Ad-PARP1 at three weeks after the adenine diet, and miR-204 agomir was intravenously injected through the tail vein for three consecutive days. miR-204 expression was determined by qRT-PCR (**h**). (n = 5 per group). The expression of osteogenic genes (OPN, ColIA1, OC, and Runx2) was analyzed by western blot (**i**). (n = 5 per group). The vascular calcification was detected by von Kossa staining (**j**) and the calcium quantification (**k**). (n = 10–12 per group). Scale bar, 100 μm. **l, m** The aortic rings (**l**) or rVSMCs (**m**) were transfected with control or miR-204 agomir together with Ad-Null or Ad-PARP1, and exposed to the osteogenic medium for 14 days. The calcification was determined by the calcium assay respectively (n = 5 per group). Statistical significance was assessed using one-way ANOVA for multiple comparison and is presented as follows: **$P$ < 0.01, #$P$ < 0.05, ##$P$ < 0.01, &$P$ < 0.05 and &&$P$ < 0.01. All values are means ± S.D. Source data are provided as a Source Data file

Runx2-targeting miRNAs. We identified several conserved miRNAs, such as miR-23a, miR-30c, miR-137, miR-204, miR-205, miR-217, miR-218, miR-221, miR-338, and miR-445 (Fig. 5c), and performed qRT-PCR on calcifying VSMCs. While the expression of miR-30c, miR-204, miR-221, and miR-338 was inhibited by high Pi concentration, only miR-204 could be reversed by the inhibitor PJ34 (Fig. 5d), which is highly conserved among the rat, mouse and human genomes (Fig. 5e). Then, we determined whether miR-204 mediated the regulation of PARP1 through Runx2. We first performed a luciferase assay after manipulating miR-204 levels. Under high Pi stimulation, the miR-204 antagomir efficiently antagonized the decreased luciferase activity of the Runx2 3′UTR due to PARP1 inhibitor (3AB and PJ34) or PARP1 deficiency (Fig. 5f). Due to the dependence

of miRNA function on the association of the miRNA/mRNA complex with Argonaute proteins to form an miRNA-induced silencing complex, we performed RNA-binding protein immunoprecipitation (RIP) using an anti-Argonaute-2 antibody and found that Pi treatment led to a reduction in the enrichment of miR-204 and Runx2 mRNA from Argonaute complexes, which was partly reversed by PJ34 (Fig. 5g). Taken together, Pi-activated PARP1 suppresses miR-204 expression and the complex formation, thus relieving the translational repression on Runx2.

Finally, to explore whether miR-204 is truly involved in vascular calcification, we injected CRF rats with a miR-204 agomir. Successful miRNA interference was confirmed by qRT-PCR (Fig. 5h), along with the serum biochemical parameters (Supplementary Table 5). Western blot analysis revealed that

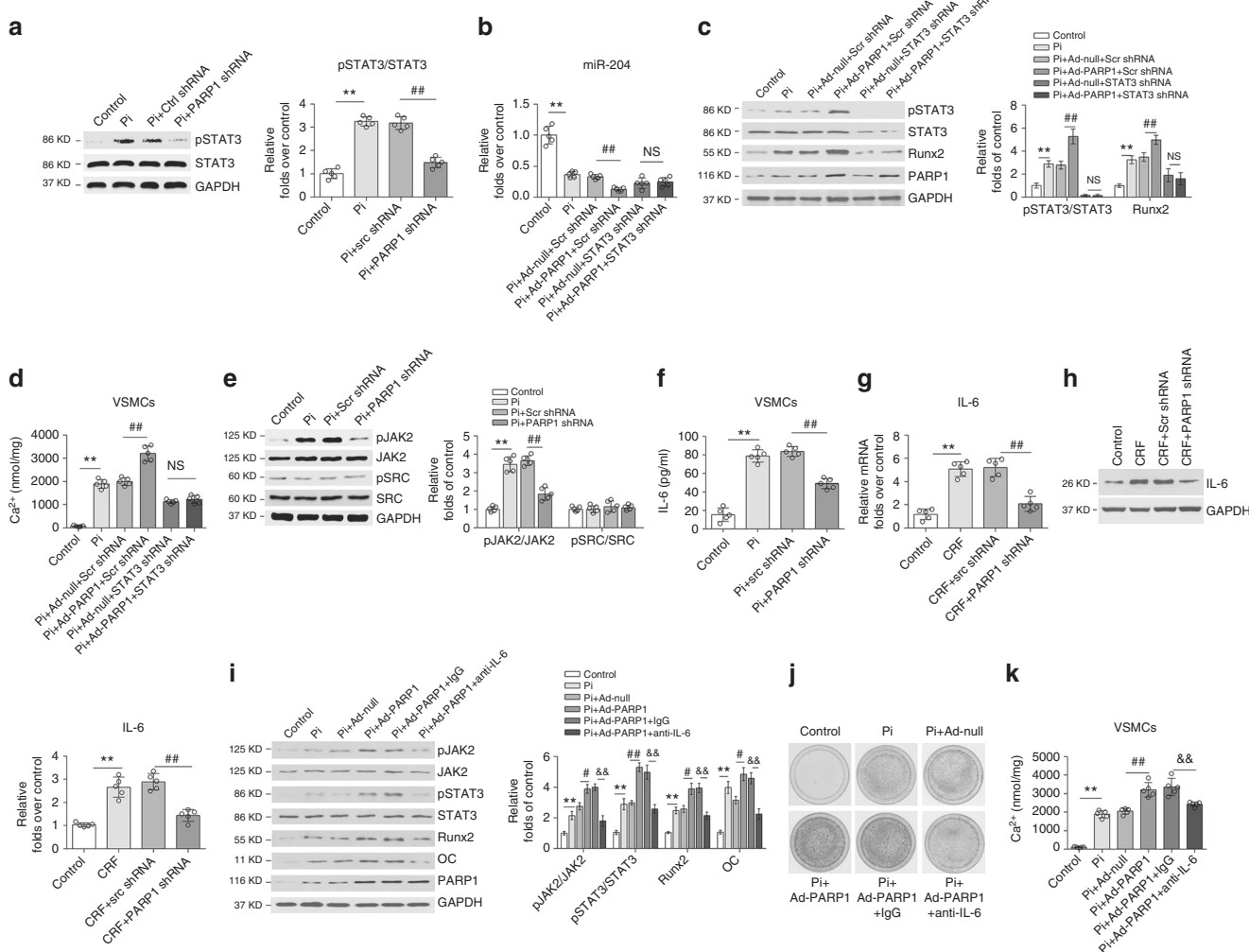

**Fig. 6** PARP1 regulates miR-204 expression through the IL-6/JAK2/STAT3 pathway. **a** Rat VSMCs was pre-infected with Scr shRNA or PARP1 shRNA, and then exposed to high Pi for 3 days. The phosphorylation of STAT3 (pSTAT3) and STAT3 were determined by western blot. ($n = 5$ per group). **b–d**, VSMCs were infected with Scr shRNA or STAT3 shRNA, along with Ad-Null or Ad-PARP1, and then incubated with osteogenic media for 14 days. MiR-204 expression was determined by qRT-PCR (**b**). The expression of pSTAT3/STAT3 and Runx2 was analyzed by western blot (**c**), and the calcium deposition was quantified by the calcium assay (**d**). ($n = 5$ per group). **e** The phosphorylation of JAK2 and SRC, and their total proteins in calcified VSMCs infected with Scr shRNA or PARP1 shRNA were determined by western blot. **f** The level of IL-6 in the supernatant of calcified VSMCs was detected by ELISA assay. **g**, **h** The mRNA (**g**) and protein (**h**) levels of IL-6 in the abdominal arteries of CRF rats inoculated with Scr shRNA or PARP1 shRNA were determined by qRT-PCR and western blot. ($n = 5$ per group). **i–k** Rat VSMCs were pre-infected with Ad-Null or Ad-PARP1, and then treated with neutralizing antibodies against IL-6 in the osteogenic media for 14 days. The levels of pJAK2/JAK2, pSTAT3/STAT3 and osteogenic markers (OC and Runx2) were determined by western blot (**i**), and the calcification was detected by Alizarin red S staining (**j**) and calcium quantification (**k**). ($n = 5$ per group). Statistical significance was assessed using one-way ANOVA followed by for multiple comparison and is presented as follows: NS: no significance, \*\*$P < 0.01$, #$P < 0.05$, ##$P < 0.01$, &$P < 0.05$ and &&$P < 0.01$. All values are means ± S.D. Source data are provided as a Source Data file

in vivo application of the miR-204 mimic repressed Runx2 expression and its downstream signaling (OPN, OCN, Col1A1) compared to Ad-PARP1 (Fig. 5i). Accompanying the decrease in gene expression, calcium deposition in vessels was also down-regulated (Fig. 5j, k and Supplementary Fig. 12f). This observation was reinforced by transfection of the miR-204 agomir into high-Pi-stimulated cultured artery rings and VSMCs. The miR-204 mimic significantly antagonized the pro-calcifying effect of PARP1 on VSMCs (Fig. 5l, m). Based on these findings, we concluded that PARP1 activation leads to a decrease in miR-204 and the subsequent release of translational repression of Runx2 mRNA, resulting in increased Runx2 protein levels and vascular calcification.

**PARP1 represses miR-204 expression via IL-6/STAT3 pathway.** STAT3 activation has been reported to suppress miR-204 expression in endometrial carcinoma cells[38]. We found that STAT3 phosphorylation and activation (increased p-STAT3/ STAT3 ratio) were substantially increased in response to high Pi concentrations and were prevented by PARP1 knockdown or PJ34 (Fig. 6a and Supplementary Fig. 9a), indicating a role for pSTAT3 in miR-204 regulation. We then silenced STAT3 via shRNA in calcified VSMCs and found that STAT3 knockdown abolished the downregulation of miR-204 mediated by PARP1 (Fig. 6b). Accordingly, the increased expression of Runx2 and osteogenic genes by PARP1, as well as calcium deposition, were all suppressed by STAT3 silencing in calcified VSMCs (Fig. 6c, d).

Both JAK2 and Src kinases are well-known tyrosine kinases that are responsible for phosphorylating and activating STAT3. We next detected the activation status of JAK2 and Src in calcified VSMCs to examine the upstream signaling. As shown in Fig. 6e and Supplementary Fig. 9b, phosphorylation of JAK2 was significantly induced in VSMCs, and PARP1 inhibition partially reversed this activation. However, Src activation was not detected.

IL-6 is known to stimulate the JAK-STAT3 signaling pathway, and cells with activated PARP1 can produce high levels of IL-6[39,40]. To test the possibility that PARP1 leads to the production and subsequent autocrine activity of IL-6 in VSMCs, cultured cell supernatants were subjected to an ELISA. Although low levels of IL-6 were detected in untreated cells, PARP1 produced a notable increase in IL-6 abundance in calcified VSMCs. Furthermore, PARP1 knockdown suppressed this upregulation (Fig. 6f). Moreover, PARP1 deficiency decreased, but PARP1 overexpression increased, the mRNA and protein levels of IL-6 in arteries of CRF rats in vivo (Fig. 6g, h and Supplementary Fig. 10). Then, we administered neutralizing antibodies against IL-6, and found that combined with blockade of STAT3 and JAK2 activation, the PARP1-enhanced calcification was nullified, as evidenced by decreased Runx2 expression and calcium content (Fig. 6i–k). In contrast, control antibodies had no effect. Together, these results provide strong evidence that autocrine IL-6 acts as a bridge between PARP1 and the JAK2-STAT3-miR-204-Runx2 pathway.

## Discussion

Our results demonstrate that PARP1 may act as an endogenous regulator of vascular calcification. Aberrant activation of PARP1 led to the osteogenic transition of VSMCs, increased mineralization-regulating proteins and subsequent calcification. PARP1 increased Runx2 protein levels by relieving the translational repression of miR-204. These data identify Runx2 as a deleterious calcifying mediator of catastrophic PARP1 activation.

PARP1 functions at the centre of cellular stress responses, where it can be activated by various stimuli and in return determines cellular fate through multiple regulatory mechanisms[11]. A growing body of evidence suggests that PARP1 is relevant to vascular pathophysiology, including vascular fibrosis[41]. However, no studies have linked PARP1 to vascular calcification. Here, we provide evidence that, in the presence of extensive ROS caused by high Pi concentrations, PARP1 was highly activated. Calcium deposition exhibited a positive correlation with PARP1 activity in arteries from CRF patients. Although our data support a role for oxidative stress in PARP1 activation, other mechanisms (DNA damage, inflammation, etc.) might also be implicated in the process. PARP1 can trigger DNA repair and promote cell survival when stress is light or moderate, but it aggravates cell death when the stress is excessive[42]. Persistent DNA damage signaling associated with cellular senescence also involves VSMC calcification[43]. So excessive DNA damage might also participate in PARP1 activation during calcification, but further study is required.

Vascular calcification is an active cell-mediated process involving VSMC apoptosis, vesicle release, a shift in the balance of inhibitors and promoters, and VSMC transdifferentiation from a contractile to an osteochondrogenic phenotype. In our study, PARP1 could indeed promote the osteogenic transition of VSMCs and calcification in rats and humans via Runx2, which was not dependent on transcriptional activity. Rather, it relied on miR-204 repression at the post-transcriptional level. Activated PARP1 suppressed miR-204 expression and function, thus relieving the translational repression of Runx2. Several lines of evidence indicate that pSTAT3 is responsible for miR-204 suppression[44]. Our results show that PARP1 could activate STAT3

(pSTAT3) in cultured VSMCs after stimulation with high Pi levels, which contributed to miR-204 downregulation. MiR-204 is encoded within intron 6 of the TRPM3 gene (transient receptor potential melastatin 3) (Supplementary Fig. 11a). PARP1 was found to co-regulate TRPM3 in the same manner as miR-204, suggesting that PARP1 downregulates miR-204 by suppressing its transcription (Supplementary Fig. 11b). Nonetheless, we silenced TRPM3 and discovered that TRPM3 did not disrupt calcification (Supplementary Fig. 11c, d).

IL-6/JAK2/STAT3 signaling was found to mediate the regulation of PARP1 by miR-204 in VSMC calcification, but how PARP1 regulates IL-6 remains unknown. NF-κB has been implicated in promoting high phosphate-induced VSMC calcification[45–47]. Inhibition of NF-κB activity within SMCs reduces arterial medial calcification in mice with CKD[48]. We and other researchers have verified NF-κB as a substrate of PARP1-mediated poly(ADP-ribosyl)ation, which then enhances NF-κB-dependent promoter activity[40,49,50]. And IL-6 is a classical direct target for activated NF-κB[39,49,51]. Interestingly, our results showed that PARP1 deficiency decreased, but PARP1 over-expression increased, the mRNA and protein levels of IL-6 in arteries of CRF rats in vivo, suggesting that PARP1 possibly upregulates IL-6 expression via promoting NF-κB transactivation during VSMC calcification.

The VSMC osteogenic phenotype is characterized by the induction of osteochondrogenic transdifferentiation markers and inhibition of smooth muscle cell lineage markers. DNA binding sites of Runx2 have been identified in the Col I, OC, and OPN genes, and Runx2 directly induces the expression of these genes or activates their promoters in osteoblasts[52–54]. Consistently, we found that increased osteogenic gene expression (OPN, OCN, CollA1) by PARP1 was dependent on Runx2. However, the mechanism by which PARP1 downregulates VSMC markers under high Pi stimulation is not fully understood. A recent study found that a direct interaction of Runx2 with SRF, which disrupts the formation of myocardin and the SRF complex, mediates the effect of Runx2 on smooth muscle cell marker gene expression[55]. It is therefore rational to speculate that elevation of Runx2 levels by PARP1 results in increased interactions with SRF, leading to suppressed expression of contractile genes, although this hypothesis requires further research.

In conclusion, this study provides evidence that PARP1 acts as a critical mediator of CRF-related vascular calcification. Under physiological conditions, VSMCs express a trace level of poly(ADP-ribosyl)ation. When phosphorus metabolism disorders occur, activated PARP1 promotes vascular calcification. Our findings provide an insight on the prevention and treatment of a variety of cardiovascular diseases related to vascular calcification.

## Methods

**Reagents, chemicals, and antibodies**. Adenine (Cat#A8626), βGP (Cat#G9422), L-ascorbic acid (Cat#A4544), dexamethasone (Cat#D4902), Alizarin Red S (Cat#A5533), Pluronic F-127 (Cat#P2443) and 3AB (Cat#A0788) were purchased from Millipore-Sigma (MO, USA). PJ34 (Cat#ALX-270-289-M025) was obtained from Enzo Life Sciences (NY, USA). The PARP universal colourimetric assay kit and antibody against PAR (Cat#4335-MC-100; 1:1000 for WB; 1:100 for IF) were purchased from Trevigen (MD, USA). Antibodies against PARP1 (Cat#9532; 1:1000), Runx2 (Cat#12556; 1:1000 for WB; 1:100 for IF), pSTAT3 (Cat#9145; 1:1000), STAT3 (Cat#12640; 1:1000), pJAK2 (Cat#3776; 1:1000), JAK2 (Cat#3230; 1:1000), pSRC (Cat#6943; 1:1000) and SRC (Cat#2109; 1:1000) were obtained from Cell Signaling Technology (MA, USA). Antibodies against SMA (Cat#ab8226; 1:2000 for WB; 1:100 for IF), osteopotin (Cat#ab8448; 1:500), OC (Cat#ab13420; 1:500), osterix (Cat#ab209484; 1:1000), MSX2 (Cat#ab223692; 1:1000). Sox9 (Cat#ab185966; 1:1000), GAPDH (Cat#ab8245; 1:2000) and Argonaute-2 (Cat#ab32381;1:100) were purchased from Abcam (Cambridge, UK). Antibodies against SM22 (Cat#10493-1-AP; 1:1000) and SMMHC (Cat#21404-1-AP; 1:1000) were obtained from Proteintech (IL, USA). Antibodies against Pit1 (Cat#a4117; 1:500) and Pit2 (Cat#a6739; 1:500) were purchased from Abclonal (Wuhan, China). The Col1A1 (Cat#NB600-408; 1:500) antibody was purchased from Novus

(CO, USA). TRPM3 (Cat#GTX16612; 1:500) antibody was purchased from Gen-eTex (CA, USA). The miR-204 agomir and its control were obtained from RIBOBIO (Guangzhou, China). Trypsin, Dulbecco's Modified Eagle's Medium (DMEM), and foetal bovine serum (FBS) were purchased from GIBCO (MA, USA). The luciferase assay system was obtained from Promega (WI, USA).

**Patient artery samples**. All procedures involving human samples complied with the principles outlined in the Declaration of Helsinki and were approved by the Institutional Review Board of Union Hospital, Tongji Medical College, Huazhong University of Science and Technology. Written informed consent was obtained from all participating patients or their legal guardians. A 5- to 8-mm segment of the radial artery was excised from uraemic patients who underwent an arterial venous fistula operation (CRF, $n = 17$) or from patients who underwent amputation surgery due to arm trauma (controls, $n = 9$) and had no diagnosed complications of diabetes or CKD. Samples were then dissected to remove fat and stored at $-80\,°C$ until use.

**Animal models**. All experimental protocols were approved by the Ethics Committee of Tongji Medical College, Huazhong University of Science and Technology, and were performed in accordance with relevant institutional and national guidelines and regulations. Eight-week-old male Wistar rats were randomized to feed with a standard chow diet containing 0.6% phosphate and 1.2% calcium for the control group or a specific diet containing 0.75% adenine, 0.9% phosphorus and 1.2% calcium for the CRF group[37]. Interventions were performed 3 weeks after initiation of the adenine diet. 3AB (30 mg/kg/d) was intraperitoneally injected every day. In the adenovirus-transfected group, rats were anesthetized with an intraperitoneal injection of pentobarbital sodium (40 mg/kg), and surgery was performed. The left renal artery was used as a marker, and the abdominal aorta was exposed ~2.0 cm above this point. Adenovirus (Ad-shPARP1, Ad-PARP1, Ad-shRunx2, Ad-Runx2, Ad-Scr shRNA or Ad-Null) ($5 \times 10^9$ plaque forming units) was mixed with 30% pluronic gel solution and applied to the exposed abdominal aorta. For the miR-204 intervention, the miR-204 agomir was intravenously injected through the tail vein for three consecutive days. Six weeks later, rats were euthanized, and blood was collected to measure blood urea nitrogen (BUN), creatinine (Cr), calcium, and phosphate using an autoanalyser (Hitachi, Japan). The abdominal arteries were excised for further analysis.

**Immunofluorescence analysis**. Vascular specimens were fixed in 4% formaldehyde and embedded in paraffin. Sections were stained with antibodies against PAR (1:100), SMA (1:100) or Runx2 (1:100) overnight at 4 °C and then incubated with Alexa Fluor® (Jackson ImmunoReasearch) secondary antibodies for 2 h at 37 °C. Nuclei were stained with DAPI. Negative controls without the primary antibody were routinely employed. Images were acquired using a fluorescence microscope (Olympus, Japan).

**Cell and vascular tissue culture**. Primary rat VSMCs were obtained by an explant method[41]. Briefly, medial tissues were separated from segments of rat aortas, cut into 1- to 2-mm² sections, and placed in a culture flask with DMEM supplemented with 15% FBS. Cells that migrated from the explants were grown in DMEM plus 15% FBS. Cells from passage 3 to 6 were used for experiments. The A7r5 and human aortic SMC (HASMC) cell lines were obtained from ATCC and grown in DMEM with 10% FBS. Rat aortic arteries were removed under sterile conditions. After removing the adventitia and endothelium, the vessels were cut into 2- to 3-mm rings and placed in DMEM containing 10% FBS. For viral infections, cells were plated overnight and then infected with adenoviruses.

**In vitro calcification and quantification**. VSMCs or aortic rings were cultured in osteogenic medium containing βGP (10 mM), L-ascorbic acid (0.25 mM), and dexamethasone ($10^{-8}$ mM) for 14 days; the medium was changed every other day. After washing with cold phosphate buffered saline (PBS), cells or rings were treated with 0.6 M HCl overnight at 4 °C. The calcium content in the HCl supernatant was subjected to colourimetric analysis using a Calcium Assay Kit (BioSino, Beijing, China) and normalized to the protein content. In parallel sets, ALP activity was measured in a colourimetric analysis using an ALP assay kit (BioSino) according to the manufacturer's instructions.

**Alizarin Red S or von Kossa staining**. For Alizarin red S staining, VSMCs in 3 cm² dishes were fixed in 4% formaldehyde for 10 min at room temperature, exposed to 2% Alizarin red S (Sigma) for 30 min and washed with 0.2% acetic acid. Positively stained cells showed a reddish/purple color. For von Kossa staining, vascular tissues sections were incubated with 5% silver nitrate solution for 30 min, exposed to bright light for 15 min, washed and treated with 5% sodium thiosulfate. Calcified nodules were stained brown to black.

A semiquantitative score of von Kossa staining was determined based on calcification in arterial cross section at three different levels using the following system: 0, no calcification; 1, focal calcification spots; 2, partial calcification covering 20–80% of the arterial circumference; and 3, circumferential calcification[56]. To avoid evaluation biases, von Kossa analyses were independently performed by two experienced pathologists who were blinded to the tissue information. Cases with discrepancies were jointly re-evaluated until a consensus was reached.

**PARP activity assay**. PARP activity was assayed using a universal colourimetric PARP assay kit (Trevigen) based on the incorporation of biotinylated ADP-ribose into histone proteins. Cell lysates containing 50 μg of proteins were loaded onto a 96-well plate coated with histones and biotinylated poly(ADP-ribose), allowed to incubate for 1 h, treated with strep-HRP, and read at 450 nm on a spectrophotometer.

**Western blot assays**. Western blotting was performed to detect protein levels in cells or tissues[57,58]. Proteins were extracted in RIPA lysis buffer (Santa Cruz) and measured using a BCA protein assay kit (Thermo). After denaturation and SDS-PAGE electrophoresis, separated proteins were transferred to PVDF membranes and probed with primary antibodies and secondary HRP-conjugated IgG antibodies. Chemiluminescence signals were detected by Image Lab statistical software (Bio-Rad). Uncropped scans for western blot analysis were shown in Supplementary Figs. 14–17.

**Generation of recombinant adenovirus**. An adenovirus kit, AdMax™ (Microbix, Canada), was used to generate adenovirus-based constructs according to the manufacturer's recommendations. Briefly, the recombinant shuttle plasmids were co-transfected with the genomic plasmid into HEK293 cells to produce recombinant viral particles, and viral titres were enriched by two rounds of infection in HEK293 cells. To generate adenoviruses encoding full-length human PARP1 (PubMed No. NM_001618.3), PARP1 mutants (L713F), or Runx2 (PubMed No. NM_001024630.3), Flag-PARP1 and Myc-Runx2 cDNA fragments were transferred from pcDNA3-based vectors to the shuttle plasmid pDC316. The CMV-null adenovirus was used as the negative control (Ad-Null).

To generate optimal adenoviruses expressing shRNA against PARP1, Runx2, STAT3, and TRMP3, three lines of corresponding adenoviruses were designed and constructed, and VSMCs were infected to detect the knockdown efficiency (Supplementary Fig. 13). The most efficient virus was used in the following experiments. The following optimal sequences were used: PARP1 shRNA with the target sequence 5′-GGATGATCTTCGACGTGGA-3′; Runx2 shRNA with the target sequence 5′-TGGCAGCACGCTATTAA-3′; STAT3 shRNA with the target sequence 5′-GGCTGATCATTTATATAAA-3′; and TRMP3 shRNA with the target sequence 5′-CCGTAAGCAAGTTTATGATTCTCAT-3′. A negative control adenovirus was designed to express non-targeting scrambled shRNA (Scr shRNA).

**Plasmid constructs and reporter assay**. The 2-kp promoter for the coding sequence of the rat *Runx2* gene was subcloned into the pGL3 luciferase reporter vector (Promega, USA). The ~4-kp full-length 3′UTR of *Runx2* mRNA was inserted into the pMIR-REPORT miRNA expression reporter vector (Ambion, USA)[27]. Luciferase reporter constructs were co-transfected into VSMCs with an internal control plasmid, pRL-TK (Renilla luciferase reporter plasmid, Promega), followed by the indicated stimulation. Then, cells were harvested and lysed, and the luciferase activity was determined with the Dual Luciferase Reporter Assay Kit (Promega, USA) according to the manufacturer's instructions.

**qRT-PCR analysis of mRNA and miRNA**. Total RNA was extracted using TRIzol (Takara, Japan). For mRNA quantification, a PrimeScript™RT reagent Kit (Takara) was used to prepare the cDNA. Real-time PCR was performed with SYBR Green (Bio-Rad, USA) on a Bio-Rad CFX-96 real-time system. The relative mRNA levels were determined by normalizing to the 18S rRNA levels. The qPCR primer pairs of Runx2 (Cat#QT01300208), PIT1 (Cat#QT00179627), PIT2 (Cat#QT00384573) and 18sRNA (Cat#QT02589300) were obtained from QIAGEN (Germany). For miRNA quantification, qRT-PCR was performed using the Bulge-Loop™ miRNA qRT-PCR Primer (RiboBio, China) following the manufacturer's protocols. U6 was detected as the internal control. The qPCR primer pairs of miR-23a, miR-30c, miR-137, miR-204, miR-205, miR-217, miR-218, miR-221, miR-338, and miR-445 were obtained from RIBOBIO (China).

**RNA-binding protein immunoprecipitation (RIP)**. RIP was performed according to the RIP Kit (Millipore). VSMCs were lysed in RIP lysis buffer, incubated with magnetic beads conjugated to anti-Argonaute-2 antibody and then rotated at 4 °C overnight. After washing, proteinase K was added to digest the proteins. RNA was purified by phenol chloroform extraction, and the purified RNA was analysed by qRT-PCR.

**Statistical analysis**. Values are shown as the means ± S.D. of at least three independent experiments. Normality of data distribution was assessed by the Shapiro–Wilk test prior to the application of parametric tests. For non-normally distributed data, nonparametric tests were used to analyze statistical differences. For comparisons between two groups, significance was determined using Student's *t*-test or nonparametric Mann–Whitney test. For comparisons among multiple groups, ANOVA followed by post hoc Bonferroni test or nonparametric

Kruskal–Wallis test followed by the Dunn's post hoc test. An F test (two groups) or Brown–Forsythe test (multiple groups) was used to determine difference in variances for t-test and ANOVA, respectively. The statistical significance of correlations was determined by Pearson's correlation coefficient analysis. Significant differences are indicated by * or # ($P < 0.05$), and very significant differences are indicated by ** or ## ($P < 0.01$). All statistical analyses were performed using SPSS software (version 22.0, SPSS Inc).

**Reporting summary**. Further information on experimental design is available in the Nature Research Reporting Summary linked to this article.

## Data availability

The data that support the findings of this study are available within the article and its Supplementary Information or from the corresponding authors on reasonable request. The source data underlying Figs. 4a, 4e, 5d, 5i, 6c, and 6i and Supplementary Figs. 5b, 5d, 6b and 7a are provided as a Source Data file.

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

## Acknowledgements

This work was supported by the National Natural Science Foundation of China (81170239), Major key technology research project of Science and Technology Department in Hubei Province (2016ACA151), Key projects of Huazhong University of Science and Technology (2016JCTD107), and the Ministry of Science and Technology of China (2016 YFA0101100). We thank Dr. Xiaomin Zhang (Department of Occupational and Environmental Health and Ministry of Education, Key Lab for Environment and Health, School of Public Health, Tongji Medical College, Huazhong University of Science and Technology) for rechecking the statistical analysis.

## Author contributions

C.W., W.J.X., and J.A. performed most of the experiments. M.L.L., F.X.Z., and Q.S.T. provided technical support, and contributed to experimental plans. Y.Q.L. provided related human samples. C.W., and W.J.X. wrote the manuscript. C.W. and K.H. conceived and supervised the study. All authors contributed to experimental design and edited the manuscript.

## Additional information

**Competing interests:** The authors declare no competing interests.

