## [Peer Review File · Nature Communications]

Reviewers' comments:

Reviewer #1 (Remarks to the Author):

In this work, the authors investigate a regulatory pathway of Cbfa1/Runx2 in vascular calcification. Runx2 is known as a master regulator of VSMC-calcification. The authors observed, that PARP1 is a key regulator of Runx2 expression by regulating Il6 secretion, subsequent activation of STAT3, miR 204 suppression and ultimately de-inhibition of Runx2 expression. This is a very interesting study that elucidates an important and understudied process, namely the signalling pathways of VSMC-calcification.

I have noted some points, that would to my understanding significantly improve the manuscript.

- How was calcification quantified in the patients? Also, a different origin of the artery specimens is noted between groups, but in Fig. 1 A is described as radial artery? A different origin of the arterial tree may bias the results. As this problem is understandable in the use of human tissue, this needs at least be discussed in a limitation section.
- Fig. 1 E and F, a higher magnification of the fluorescence pictures is required for localization of PARP (activity) in the sections. With the current pictures it is not possible to interpret possible autofluorescence or proper localization. Also, the localization may be different: at 7 days in intima and adventitia, at 10 days media?
- Fig. 3 + 4. Is overexpression of PARP1 alone sufficient to induce osteogenic markers expression?
- Fig. 4 B – a better quality of runx2 pictures will ease interpretations, there is faint staining only in the pictures also with high nuclei staining.
- The method section is too rudimentary. Eg how was abdominal aorta silencing conducted in vivo. Such advanced methodology should be described. Also, the injection of miR-204 agomir is not sufficiently explained.
- What does contraction and calcification mean in fig.3 ? It is not easily interpretable.
- An important proof of concept experiment would be overexpressing runx2 in the presence of PARP inhibition.
- Fig. 4 C-F. the group CRF/Pi + Runx2 shRNA is missing as knockdown of Runx2 is protective beyond the effects of PARP1. This should be at least be addressed in vitro.

A suitable way to underscore a direct dependence would be to overexpress PARP1 with and without silencing of RUNX2 without phosphate treatment.

- In the tables at the CRF+Ad-PARP1+Scr shRNA section the CRF group shows BUN/Pi levels like control rats. The table should be rechecked for correctness. What is Cr?

- Fig. 6. Should be further explored, especially the effect of STAT3 silencing on RUNX2 protein levels and calcification.

Possibly: JAK2 inhibition and mir-204 levels, RUNX2 protein abundance and calcification

& Effect of anti-IL6 antibody on phospho JAK2, RUNX2 protein levels.

It should be at least discussed in more detail how PARP1 regulates IL-6 levels.

- The authors rely on rat VSMCs and even the A7r5 cell line. This may hinder translation to the human and should be mentioned as limitation.

- Along these lines: Fig. 5 The miR targeting runx2 are analysed for the rat sequence. Is the miR-204 target sequence conserved in RUNX2 gene in other species, especially in humans?

- Statistics: The authors use testing of homogeneity of variances for proper statistical testing. But then only Newman-keuls test is used, which to my knowledge requires homogeneity of variances. What was used if homogeneity of variances was not given in the data? How was correlation testing performed?

- In general, several grammar errors were observed that should be corrected. Some of the phrasing should be refined as it may lead to (unintended) wrong interpretations. The manuscript would strongly benefit by proofreading from a native English speaker.

- "VSMC differentiation" is not a perfectly suitable term and should be replaced with transdifferentiation.

- In the discussion another second mechanism of PARP induced SMC marker repression is discussed, which is hard to follow. "Our another experiment has verified that PARP1 can bind with myocardin and SRF, decreases the complex DNA binding and thus suppresses SMC contractile genes (SMA, SM22 α and SMMHC), which leading us to suspect that PARP1 might disrupt Runx2 and myocardin binding to regulate SMC marker genes". Where is that data shown? Or is this lacking a citation?

- A Major Resources Table mentioned in the manuscript was not clearly visible.

- The supplemental figure 4 is rather cryptic to me as it lacks a proper figure legend and is not cited in the text. I'm assuming it is the densitometry of the data shown in the main figures?

Reviewer #2 (Remarks to the Author):

The authors showed that PARP1 activity increased during the vascular calcification associated with enhanced generation of ROS. Lack of PARP1 decreased vascular calcification level and PARP1 overexpression enhanced vascular calcification. PARP1 aggravates vascular calcification via suppressing miRNA-204 that acts on Runx2 through IL-6/STAT3 activation through JAK2 modulation. The role of PARP1 in vascular calcification was not previously studied in detail and the finding has an impact on the mechanism of vascular calcification and also on the therapeutic point of view. However, the manuscript needs following improvements to clarify the significances of the results.

- 1) The Methods lack the description of shRNA and adenovirus transfection and vector structure details and the information should be described. Two kinds of shRNA should be used for the key experiments to avoid off-target effects.
- 2) Von Kossa staining of the arteries should be scored, quantified and statistical differences should be shown (Fig. 1e, Fig. 2H, Fig. 4D, Fig. 5J). Fig. numbering should be unified.
- 3) In the experiments of Fig. 3, Fig. 4, Fig. 5 and Fig. 6, which used PARP1 shRNA or overexpression, the changes in PARP-1 level should be analyzed. Fig. 6F needs to show the changes in SRC mRNA or protein level.
- 4) In Fig. 3B, Fig. 5F/G/H, and Fig. 6C, control should also be presented with standard error bars.
- 5) Fig. 5C, the length of 3'-UTR analyzed and positions of miR sequences should be indicated.
- 6) The control is missing in Fig. 6G and should be added.
- 7) Lines 126: "Bio-Red" should be corrected to "Bio-Rad".
- 8) Lines 134-138: data which used RIP method was not appeared and should be removed.
- 9) Lines 155-158: "PARP1" should be changed to "PARP", because the used method does not distinguish PARP family proteins.
- 10) Line 158: "strongly associated" should be toned down to "associated".
- 11) Line 195: " PARP1" should be corrected to "shRNA of PARP1" or similar term.
- 12) Lines 283-285: the sentences should be edited.
- 13) Lines 303 and 310 should also be corrected.
- 14) Supple Fig. 1B legend: "was incubation" should be corrected to "was incubated".

Reviewer #3 (Remarks to the Author):

The authors have studied the role of PARP1 in vascular calcification in vivo in rats and in vitro in primary rat vascular smooth muscle cell (VSMC) culture through over expression and inhibition/knock down of PARP1. They convincingly demonstrate a procalcific role for PARP-1 in uremic arterial medial calcification using the adenine rat model, along with a mechanistic study that point to an involvement of the IL-6/STAT3 pathway in regulation of Runx2 via miR-204.

There are several major and minor concerns that should be addressed:

Major issues:

1. Human specimen concerns: PARP1 is known to be critical in regulation of atherosclerosis. Using internal mammary artery specimens collected from patients that have undergone coronary artery bypass surgery as the “normal” control for radial arteries from CRF patients is not appropriate. Were these arteries calcified and their osteogenic gene expression levels tested? Also, were patients provided with appropriately informed consent for the tissues collected and used in this study? I suggest that in Figure 1 and legend, remove radial artery, since CRF and control arteries are not from the same source. Figure 1c, was p value <0.05 or 0.01? It's not consistent in legend & the result section. Figure 1e, f, need better images need quantitation with normalization. What is PAR staining, cells or matrix?
2. High phosphate models (hyperphosphatemia in vivo & high phosphate in vitro and ex vitro) were used throughout the study. It is not very convincing to exclude role of Pi/transporter by solely comparing the expression levels in the high Pi condition, as levels may be maximal under these conditions. Manipulation of PiT expression along with PARP1 delivery is required for the conclusion.
3. Replications: what are the biological sample replications for Western blot analysis shown in Figures 3 through 6? Quantitative analysis should be performed for the data that used to draw key conclusions, such as Runx2 levels in the shRNA study shown in Figure 4 and the agomiR-204 study shown in Figure 5I, and the p-STAT3/STAT3 ratio shown in Figures 6A and 6B.

4. Did Runx2 expression level change in CRF PARP1 shRNA and Ad-PARP1 samples? Were CRF Ad-PARP1 arteries high in IL-6 while shRNA reduced it?

5. Online supplement figure 1 was generated from cultured VSMCs. It is not clear why ROS in uremia was described at the start of the result section, page 8. In general, the paper focuses a lot on ROS, yet Intro does not do a good job of making the connection to CRF and calcification

6. PARP1 shRNA and expression constructs were not described and it is unclear how these genetic interferences were delivered, e.g., via surgery? Ad-Null and Ad-PARP1 need to be defined. The supplementary table describing reagents was not found.

7. Figure 3, did shRNA or Ad-PARP1 transfection to VSMCs alter phosphate uptake by these cells? To conclude that PARP1 regulate VSMC phenotype switch, a quantitative measure should be provided.

8. It is difficult to see the staining for DAPI in some of the images shown in Figure 1f, and Runx2 in Figure 4B. Better IF images and higher magnification need to be provided.

9. Some of the von Kossa images shown in Figure 5J do not reflect the quantitative findings.

10. The authors need to clarify if the staining is von kossa or Alizarin red in Figure 2D

11. Citation problems: The 2004 reference #16 is not an appropriate citation for the current understanding of the role of Runx2 in vascular calcification. Runx2 has been shown to be critical for vascular calcification in several recent experimental animal models (Sun et al, Circ Res 2012; Lin et al AJP 2015; Lin et al CVR 2016) . These recent studies should be cited, as they provide important impetus for the focus on Runx2 in the present studies, especially in the context of abnormal mineral metabolism (Lin et al AJP 2015).

12. (PARP1) has been previously reported to be involved in calcification (Nagy E et al, Increased transcript level of poly(ADP-ribose) polymerase (PARP-1) in human tricuspid compared with bicuspid

aortic valves correlates with the stenosis severity. Biochemical and biophysical research communications. 2012;420:671-5.) This paper is not mentioned in the paper, but it quite relevant.

Minor Concerns:

Typos throughout the paper that require correction

Some sentences require rewording for clarity (Ex lines 149-151, 309-311)

Reviewer #1 (Remarks to the Author):

In this work, the authors investigate a regulatory pathway of Cbfa1/Runx2 in vascular calcification. Runx2 is known as a master regulator of VSMC-calcification. The authors observed, that PARP1 is a key regulator of Runx2 expression by regulating Il6 secretion, subsequent activation of STAT3, miR 204 suppression and ultimately de-inhibition of Runx2 expression. This is a very interesting study that elucidates an important and understudied process, namely the signaling pathways of VSMC-calcification.

I have noted some points, that would to my understanding significantly improve the manuscript.

1. How was calcification quantified in the patients? Also, a different origin of the artery specimens is noted between groups, but in Fig. 1 A is described as radial artery? A different origin of the arterial tree may bias the results. As this problem is understandable in the use of human tissue, this needs at least be discussed in a limitation section.

Response: Part of the arteries from patients was incubated in 0.6M HCl at 4°C overnight. After collecting the supernatant, the remaining tissues were then dissolved in 0.1M NaOH and 0.1% SDS for protein concentration analysis. The calcium content in the supernatant was colorimetrically analyzed by a Calcium Assay Kit (BioSino) and normalized to protein content. We feel sorry for the wrongly labelled title of densitometry data in Fig.1a. We firstly chose radial arteries and internal mammary arteries according to the paper ¹. Considering the different origin of the arterial tree, we collected radial arteries from patients who underwent amputation surgery due to arm trauma as controls, without diagnosed complications of diabetes and chronic kidney diseases. All samples were obtained with the agreement of the patients and approved by the Institutional Review Board of Union Hospital, Tongji Medical College, Huazhong University of Science and Technology. The Poly(ADP-ribosylation) levels and calcium content in radial arteries were quantified, which showed an increase in both PARP activity and calcium deposition in arteries of CRF compared with that in controls.

2. Fig. 1 E and F, a higher magnification of the fluorescence pictures is required for localization of PARP (activity) in the sections. With the current pictures it is not possible to interpret possible autofluorescence or proper localization. Also, the localization may be different: at 7 days in intima and adventitia, at 10 days media?

Response: We feel really for the bad magnification of the pictures. The different localization was largely due to the bad immunofluorescence. We have selected new related antibodies and performed the immunofluorescence assay again. The pictures were all well improved in **Fig.1d** and **e**.

3. Fig. 3 + 4. Is overexpression of PARP1 alone sufficient to induce osteogenic markers expression?

Response: No, PARP1 alone is not sufficient to induce osteogenic genes expression (**Supplementary Fig. 6a**). We suppose that may be attributed to the inactivation of PARP1 without high Pi stimulus.

4. Fig. 4 B – a better quality of runx2 pictures will ease interpretations, there is faint staining only in the pictures also with high nuclei staining.

Response: We have improved the immunofluorescence pictures in **Fig.4b**.

5. The method section is too rudimentary. Eg how was abdominal aorta silencing conducted in vivo. Such advanced methodology should be described. Also, the injection of miR-204 agomir is not sufficiently explained.

Response: We are sorry for the method section. We have corrected and added the detailed steps or protocols in the method section.

6. What does contraction and calcification mean in fig.3? It is not easily interpretable.

Response: Pi-induced vascular calcification has been reported to be an adaptive VSMC transition from a contractile to an osteogenic/chondrogenic phenotype, characterized by the upregulation of osteogenic gene expression (osteopontin (OPN), osteocalcin (OC) and collagen IA1 (ColIA1)), and the simultaneous downregulation of VSMC contractile gene expression (Alpha -Smooth Muscle Actin (α -SMA), Smooth Muscle Protein 22 (SM22) and Myosin Heavy Chain 11 (SMMHC))^{2,3,4}. Therefore, we used contraction to indicate the VSMC contractile phenotype, and calcification to indicate the VSMC osteogenic/chondrogenic phenotype. As shown in **Fig.3**, our data verified that PARP1 could promote VSMC osteogenic transition.

7. An important proof of concept experiment would be overexpressing runx2 in the presence of PARP inhibition.

Response: Yes, we have overexpressed Runx2 in the presence of PARP inhibitor PJ34 *in vitro*. As shown in **Supplementary Fig. 7**, ectopic Runx2 was able to antagonize the protective effects of PJ34 on Pi-induced VSMC calcification, suggesting that Runx2 possibly mediates the effects of PARP1.

8. Fig. 4 C-F. the group CRF/Pi + Runx2 shRNA is missing as knockdown of Runx2 is protective beyond the effects of PARP1. This should be at least be addressed in vitro.

A suitable way to underscore a direct dependence would be to overexpress PARP1 with and without silencing of RUNX2 without phosphate treatment.

Response: In the third question, we found that PARP1 alone was unable to induce osteogenic genes expression. We supposed that might be attributed to the inactivation of PARP1 without high Pi stimulus. The PARP1 L713F is constitutively active in cells^{5,6,7}. As shown in supplementary Fig.6, we selected PARP1-WT and PARP1-L713F mutant adenovirus to avoid the primary activation of PARP1 by phosphate. VSMCs were infected with Ad-Scr shRNA or Ad-Runx2 shRNA, together with PARP1-WT or PARP1-L713F mutant adenovirus as indicated. Proteins were then extracted for genes expression by western blot and the calcium content was calculated in parallel. Interestingly, WT could not, but L713F could induce the osteogenic genes expressions as well as the calcium deposition. The increased calcium deposition was much less than the high phosphate-induced calcification, likely due to the lack of specific disturbed calcium/phosphate environment. When Runx2 was depleted, the effects of L713F was nearly abolished (**Supplementary Fig. 6b, c**). In conclusion, we suggest a direct dependence of PARP1 on expression of Runx2 without phosphate treatment, but phosphate exposure is essential for PARP1/Runx2-mediated vascular calcification. Besides, we have supplemented the group CRF+Ad-Null+Runx2 shRNA in **Fig. 4c-g**.

9. In the tables at the CRF+Ad-PARP1+Scr shRNA section the CRF group shows BUN/Pi levels like control rats. The table should be rechecked for correctness. What is Cr?

Response: We feel sorry for the mistakes. After check, we wrongly put the data of control rats in the CRF group. We have corrected them. Cr is the abbreviation of creatinine, and BUN is the abbreviation of urea nitrogen, both of which indicates the renal function.

10. Fig. 6. Should be further explored, especially the effect of STAT3 silencing on RUNX2 protein levels and calcification.

Possibly: JAK2 inhibition and mir-204 levels, RUNX2 protein abundance and calcification
& Effect of anti-IL6 antibody on phospho JAK2, RUNX2 protein levels.

It should be at least discussed in more detail how PARP1 regulates IL-6 levels.

Response: Yes, we have further detected the effects of STAT3 silencing in VSMCs on miR-204 level, Runx2 abundance and the calcification. As expected in **Fig. 6b-d**, STAT3 silencing attenuated PARP1-enhanced miR-204 and Runx2 expressions, together with the decreased calcium deposition. Then using neutralizing antibodies to IL-6, the portion of PARP1-enhanced pJAK2, pSTAT3, Runx2 and calcification was mostly nullified (**Fig. 6i-k**), whereas control antibodies had no effects, suggesting a critical role of IL-6/JAK2/STAT3/miR-204/Runx2 pathway in the regulation of VSMC calcification. Studies have verified that PARP-1 could bind to NF-kappaB and promote NF-kappaB/DNA complex formation, thereby enhancing the expression of IL-6 in cells.^{8,9,10} Moreover, we and other researchers have revealed NF-kappaB as a substrate of PARP1-mediated poly(ADP-ribosylation), which then enhanced NF-kappaB-dependent promoter activity^{9,11,12}. -Interestingly, in this research, we found that PARP1 deficiency decreased, but PARP1 overexpression increased, the RNA and protein levels of IL-6 in arteries of CRF rats *in vivo* (**Fig.6g, h** and **Supplementary Fig. 10**), all supposing that PARP1 possibly upregulates IL-6 expression via promoting NF-kappaB transactivation in VSMC calcification.

11. The authors rely on rat VSMCs and even the A7r5 cell line. This may hinder translation to the human and should be mentioned as limitation.

Response: We have applied human aortic smooth muscle cells (HASMC) to test the osteogenic transition in calcification. As shown in **Supplementary Fig. 5**, high Pi increased osteogenic genes (OPN, OCN), but decreased contractile genes (α -SMA, SMMHC) expressions in HAVSMCs, which was reversed by PARP inhibitors 3AB and PJ34, or PARP1 silencing. Furthermore, PARP inhibition or depletion could also alleviate high Pi-induced calcium deposition by alizarin red staining and calcium content assay, which suggests that the effects of PARP1 on Pi-induced VSMC calcification is highly conservative in both human and rat species.

12. Along these lines: Fig. 5 The miR targeting runx2 are analysed for the rat sequence. Is the miR-204 target sequence conserved in RUNX2 gene in other species, especially in humans?

Response: Yes, the miR-204 target sequence in Runx2 was conserved in rat, mouse and human (**Fig.4e**).

13. Statistics: The authors use testing of homogeneity of variances for proper statistical testing. But then only Newman-keuls test is used, which to my knowledge requires homogeneity of variances. What was used if homogeneity of variances was not given in the data? How was correlation testing performed?

Response: Yes, We have done it. The homogeneity of variance was assessed by the F test (two groups)

or Brown-Forsythe test (≥ 3 groups). The statistical significance of differences between two groups was analysed by Student's t-tests. Non-parametric data were analysed by Mann-Whitney U test. To compare more than two means, one-way ANOVA followed by the Newman-Keuls test was used for parametric data. Kruskal-Wallis test followed by the Dunn's post hoc test was used for nonparametric data. The statistical significance of correlations was determined by Pearson's correlation coefficient analysis.

14. In general, several grammar errors were observed that should be corrected. Some of the phrasing should be refined as it may lead to (unintended) wrong interpretations. The manuscript would strongly benefit by proofreading from a native English speaker.

Response: We have invited the agency Springer Nature Author Services to edit and proofread our article. The article was edited extensively, and we hope the revised manuscript could be acceptable for you.

15. "VSMC differentiation" is not a perfectly suitable term and should be replaced with transdifferentiation.

Response: Yes, we have replaced "transdifferentiation" to indicate VSMC osteogenic transition.

16. In the discussion another second mechanism of PARP induced SMC marker repression is discussed, which is hard to follow. "Our another experiment has verified that PARP1 can bind with myocardin and SRF, decreases the complex DNA binding and thus suppresses SMC contractile genes (SMA, SM22 and SMMHC), which leading us to suspect that PARP1 might disrupt Runx2 and myocardin binding to regulate SMC marker genes". Where is that data shown? Or is this lacking a citation?

Response: Runx2 is reported to directly regulate osteogenic genes (Col I, OC and OPN) expression as a transcriptional factor, not for the contractile genes, so the mechanism how PARP1/Runx2 regulated contractile genes was not fully understood. Previous research showed that a direct interaction of Runx2 with SRF that disrupts the formation of myocardin and SRF ternary complex mediated the effect of Runx2 on SMC marker gene expression¹³. And our unpublished study discovered that PARP1 could bind to SRF to suppress contractile genes (α -SMA, SM22, SMMHC) expressions. So we supposed that elevated Runx2 levels by PARP1 resulted in more interaction with SRF, leading to suppressed expression of contractile genes. We feel really sorry for the unclear descriptions. We have corrected this section. Since this study is being submitted, we did not provide the data in the paper.

17. A Major Resources Table mentioned in the manuscript was not clearly visible.

Response: This table mainly includes the detail information of antibodies and primers used in the paper. We have added the details in the method section.

18. The supplemental figure 4 is rather cryptic to me as it lacks a proper figure legend and is not cited in the text. I'm assuming it is the densitometry of the data shown in the main figures?

Response: Yes, it's the densitometry of the data in the main figures. We have reorganized the figures and put them in the right and suitable places for easier reading.

Reviewer #2 (Remarks to the Author):

The authors showed that PARP1 activity increased during the vascular calcification associated with enhanced generation of ROS. Lack of PARP1 decreased vascular calcification level and PARP1 overexpression enhanced vascular calcification. PARP1 aggravates vascular calcification via suppressing miRNA-204 that acts on Runx2 through IL-6/STAT3 activation through JAK2 modulation. The role of PARP1 in vascular calcification was not previously studied in detail and the finding has an impact on the mechanism of vascular calcification and also on the therapeutic point of view. However, the manuscript needs following improvements to clarify the significances of the results.

1) The Methods lack the description of shRNA and adenovirus transfection and vector structure details and the information should be described. Two kinds of shRNA should be used for the key experiments to avoid off-target effects.

Response: We have supplemented the shRNA sequences and related details in the method. We firstly constructed three lines of corresponding adenovirus, then infected VSMC to detect the knockdown efficiency, and finally used the most efficient one. To detect the key role of PARP1 in Pi-induced VSMC calcification, we used three kinds of shRNA adenovirus to infect VSMCs. All these three kinds of shRNA had beneficial effects on Pi-induced VSMCs calcification by alizarin red staining and calcium content assay. We chose the best one in doing the following experiments (**Supplementary Fig.13**).

The efficiency of PARP1, Runx2, TRPM3 and STAT3 shRNA. a-c, VSMCs were infected with adenovirus encoding three different lines of PARP1 shRNA for 48 hours, and then exposed to osteogenic media for 14 days. The protein level of PARP1 was determined by western blot assay (a). VSMCs were stained for mineralization by Alizarin red S (b), and the quantitative analysis of calcium content were detected (c). d-f, VSMCs were separately infected with Scr shRNA, three different lines of Runx2 shRNA, TRPM3 shRNA and STAT3 shRNA for 48 hours, and then the mRNA level of Runx2 (d), TRPM3(e) and STAT3 (f) were determined by qRT-PCR. (n=5 per group). Statistical significance was assessed using one-way ANOVA for multiple comparison, **p < 0.01. All values are means \pm SEM of three independent experiments.

2) Von Kossa staining of the arteries should be scored, quantified and statistical differences should be shown (Fig. 1e, Fig. 2H, Fig. 4D, Fig. 5J). Fig. numbering should be unified.

Response: Yes, we have added the statistical data of the von kossa staining in **Supplementary Fig.12** and unified the Fig. numbering.

3) In the experiments of Fig. 3, Fig. 4, Fig. 5 and Fig. 6, which used PARP1 shRNA or overexpression, the changes in PARP-1 level should be analyzed. Fig. 6F needs to show the changes in SRC mRNA or protein level.

Response: We have added the changes in PARP1 levels in all related figures. In **Fig. 6e** and **Supplementary Fig. 9b**, we have showed that PARP1 inhibition or silencing cannot alter the protein level of SRC. As shown in below, PARP1 inhibition or silencing also had no effect on the mRNA level of SRC.

Rat VSMCs was treated with PARP1 inhibitors or PARP1 shRNA, and then exposed to osteogenic media for 3 days. The mRNA level of SRC was determined by qRT-PCR. Statistical significance was assessed using one-way ANOVA for multiple comparison. All values are means \pm SEM of at least three independent experiments.

4) In Fig. 3B, Fig. 5F/G/H, and Fig. 6C, control should also be presented with standard error bars.

Response: Yes, we have revised the presentation of controls in revised **Fig. 3a**, **Fig. 5g, h**, and **Fig. 6b**.

5) Fig. 5C, the length of 3'-UTR analyzed and positions of miR sequences should be indicated.

Response: Yes, we added all the information in the **Fig. 5c**.

6) The control is missing in Fig. 6G and should be added.

Response: We have added the controls in revised **Fig.6i**.

7) Lines 126: "Bio-Red" should be corrected to "Bio-Rad".

Response: We have corrected it.

8) Lines134-138: data which used RIP method was not appeared and should be removed.

Response: The data using RIP method was shown in **Fig.5g**, which suggested that high Pi decreased, but PJ34 increased the miR-204/Runx2 3'UTR/Argonaut complex.

9) Lines 155-158: "PARP1" should be changed to "PARP", because the used method does not distinguish PARP family proteins.

Response: We have changes "PARP1" to "PARP".

10) Line 158: "strongly associated" should be toned down to "associated".

Response: We have corrected the words accordingly.

11) Line 195: " PARP1" should be corrected to "shRNA of PARP1" or similar term.

Response: We have changed "PARP1" to "PARP1 deficiency".

12) Lines 283-285: the sentences should be edited.

Response: We have corrected the sentences in the revised paper.

13) Lines 303 and 310 should also be corrected.

Response: We have corrected the words.

14) Supple Fig. 1B legend: “was incubation” should be corrected to “was incubated”.

Response: We revised accordingly.

Reviewer #3 (Remarks to the Author):

The authors have studied the role of PARP1 in vascular calcification in vivo in rats and in vitro in primary rat vascular smooth muscle cell (VSMC) culture through over expression and inhibition/knock down of PARP1. They convincingly demonstrate a procalcific role for PARP-1 in uremic arterial medial calcification using the adenine rat model, along with a mechanistic study that point to an involvement of the IL-6/STAT3 pathway in regulation of Runx2 via miR-204.

There are several major and minor concerns that should be addressed:

Major issues:

1. Human specimen concerns: PARP1 is known to be critical in regulation of atherosclerosis. Using

internal mammary artery specimens collected from patients that have undergone coronary artery bypass surgery as the “normal” control for radial arteries from CRF patients is not appropriate. Were these arteries calcified and their osteogenic gene expression levels tested? Also, were patients provided with appropriately informed consent for the tissues collected and used in this study? I suggest that in Figure 1 and legend, remove radial artery, since CRF and control arteries are not from the same source. Figure 1c, was p value <0.05 or 0.01? It's not consistent in legend & the result section. Figure 1e, f, need better images need quantitation with normalization. What is PAR staining, cells or matrix?

Response: Yes, selected patients were provided with appropriately informed consent for the tissues collected and used in this study. We firstly chose radial arteries and internal mammary arteries according to the paper¹. Considering the different origin of the arterial tree, we collected radial arteries from patients who underwent amputation surgery due to arm trauma as controls, without diagnosed complications of diabetes and chronic kidney disease. All samples were obtained with the agreement of the patients and approved by the Institutional Review Board of Union Hospital, Tongji Medical College, Huazhong University of Science and Technology. We tested the calcium content in radial arteries samples. The calcium content were significantly higher in radial arteries from CRF compared with that in controls. We feel sorry the mistakes in Fig.1c. The p value was <0.05, and we have corrected it in the legend. Better images for PAR in arteries have been supplied and quantitation in **Fig.1d** and **e**.

2. High phosphate models (hyperphosphatemia in vivo & high phosphate in vitro and ex vitro) were used throughout the study. It is not very convincing to exclude role of Pi/transporter by solely comparing the expression levels in the high Pi condition, as levels may be maximal under these conditions. Manipulation of PiT expression along with PARP1 delivery is required for the conclusion.

Response: Yes, We have found that PARP1-L713F, which is constitutively active in cells.^{5,6,7}, could induce osteogenic genes expression without phosphate treatment (**Supplementary Fig. 6**). So we selectively overexpressed PARP1 L713F mutant to avoid the effect of decreased phosphate intake when Pit expression was knock down. Ad-PARP1-L713F adenovirus treatment increased the osteogenic gene expression, but Pit1/2 silencing could not alter the Ad-PARP1-L713F-induced expression of osteogenic genes. In addition, elevated Pi content in VSMCs could directly stimulate VSMCs to undergo osteogenic transdifferentiation and matrix calcification¹⁴. We found that high Pi treatment could dramatically increase the intracellular Pi concentration in VSMCs, but PARP inhibition 3AB or PJ34, and PARP1 shRNA could not alter the Pi concentration in VSMCs exposed to Pi. All suggests that role of Pi/transporter is not involved in PARP1-related VSMCs osteogenic transdifferentiation.

Pit1/Pit2 has no effects on PARP1-mediated VSMC calcification. a, VSMCs were infected with Ad-Null, Ad-PARP1-WT or Ad-PARP1-L713F, together with Scr shRNA or Pit1/2 shRNA for 48 hours, and then incubated with osteogenic media for 14 days. The osteogenic markers (OPN, Col1A1, OC and Runx2) in VSMCs were determined by western blot. b and c, VSMCs was treated with PJ34 (10 μ M) or infected with Ad-Scr shRNA and , Ad-PARP1 shRNA with 48 hours, and then the Pi uptake was determined in EBSS medium containing 0.1 mmol/L $H_3^{32}PO_4$ (1 mCi/mL) for the indicated times, which was normalized by cellular protein content. (n=5 per group). Statistical significance was assessed using one-way ANOVA for multiple comparison and is presented as follows: NS: no significance, ** P < 0.01. All values are means \pm SEM of three independent experiments.

3. Replications: what are the biological sample replications for Western blot analysis shown in Figures 3 through 6? Quantitative analysis should be performed for the data that used to draw key conclusions, such as Runx2 levels in the shRNA study shown in Figure 4 and the agomiR-204 study shown in Figure 5I, and the p-STAT3/STAT3 ratio shown in Figures 6A and 6B.

Response: The biological samples were performed at least in triplicate independently in Fig. 3 through 6. We have added the quantitative analysis for the Western blot bands or miR-204 levels in the related figures.

4. Did Runx2 expression level change in CRF PARP1 shRNA and Ad-PARP1 samples? Were CRF Ad-PARP1 arteries high in IL-6 while shRNA reduced it?

Response: Yes, the level of Runx2 was decreased in arteries of Ad-PARP1 shRNA + CRF rats (Fig.4b), and increased in Ad-PARP1+CRF rats (Fig.4e) *in vivo*. As shown in Fig. 6g, h and Supplementary Fig.10, PARP1 deficiency decreased, but PARP1 overexpression increased IL-6 mRNA and protein levels in CRF arteries.

5. Online supplement figure 1 was generated from cultured VSMCs. It is not clear why ROS in uremia was described at the start of the result section, page 8. In general, the paper focuses a lot on ROS, yet

Intro does not do a good job of making the connection to CRF and calcification

Response: Some studies have reported increased ROS in CRF^{15, 16, 17, 18}, and ROS is considered as a classic activator for PARP1^{19, 20}, so we speculated that in CRF PARP1 may be activated and participate in the process. That's why we chose ROS at the start of the result section. We have revised the introduction for a better understanding.

6. PARP1 shRNA and expression constructs were not described and it is unclear how these genetic interferences were delivered, e.g., via surgery? Ad-Null and Ad-PARP1 need to be defined. The supplementary table describing reagents was not found.

Response: PARP1 shRNA and expression constructs were described detailedly in method section. All these genetic interferences were delivered via surgery. Rats were anesthetized with pentobarbital sodium by intraperitoneal injection (40 mg/kg). After exposure, the left renal artery was used as a marker, and the abdominal aorta was exposed ~2.0 cm above this point. Adenovirus (Ad-Null, Ad-PARP1, Scr shRNA and PARP1 shRNA) (5×10^9 plaque forming units) was mixed with 200 μ l matrix gel (Sigma) and smeared around the exposed abdominal aorta. We have added the describing reagents to the method section in this version.

7. Figure 3, did shRNA or Ad-PARP1 transfection to VSMCs alter phosphate uptake by these cells? To conclude that PARP1 regulate VSMC phenotype switch, a quantitative measure should be provided.

Response: No, we found PARP1 shRNA or inhibitors did not alter VSMC phosphate uptake (please see the related data in the second response Fig. b and c). We also performed Pi/transporter (Pit1/2) silencing experiments, which showed that Pit1/2 knockdown could not influence the effect of PARP1 on VSMC phenotype switch. The quantitative measures were provided in **Fig. 3**. We reorganized figures and put them in the right and suitable places.

8. It is difficult to see the staining for DAPI in some of the images shown in Figure 1f, and Runx2 in Figure 4B. Better IF images and higher magnification need to be provided.

Response: We have selected new antibodies and performed the immunofluorescence again. The IF pictures and higher magnification were provided in **Fig. 1e** and **4b**.

9. Some of the von Kossa images shown in Figure 5J do not reflect the quantitative findings.

Response: We are sorry the mistakes. We have wrongly put the second and the fourth pictures. We have corrected them.

10. The authors need to clarify if the staining is von kossa or Alizarin red in Figure 2D

Response: The staining in Figure.2d is Alizarin red S staining. We have added the clarification in the legends.

11. Citation problems: The 2004 reference #16 is not an appropriate citation for the current understanding of the role of Runx2 in vascular calcification. Runx2 has been shown to be critical for vascular calcification in several recent experimental animal models (Sun et al, Circ Res 2012; Lin et al AJP 2015; Lin et al CVR 2016). These recent studies should be cited, as they provide important impetus for the focus on Runx2 in the present studies, especially in the context of abnormal mineral metabolism (Lin et al AJP 2015).

Response: We are sorry for the inappropriate citation. We have corrected and cited the above studies.

12. (PARP1) has been previously reported to be involved in calcification (Nagy E et al, Increased transcript level of poly(ADP-ribose) polymerase (PARP-1) in human tricuspid compared with bicuspid aortic valves correlates with the stenosis severity. *Biochemical and biophysical research communications*. 2012;420:671-5.) This paper is not mentioned in the paper, but it quite relevant.

Response: Yes, we added the reference and compared the difference in the introduction section.

Minor Concerns:

Typos throughout the paper that require correction

Some sentences require rewording for clarity (Ex lines 149-151, 309-311)

Response: We have corrected the typos and some wrong sentences, and revised our paper in the agency Springer Nature Author Services.

Reference

1. Du Y, et al. Cartilage oligomeric matrix protein inhibits vascular smooth muscle calcification by interacting with bone morphogenetic protein-2. *Circ Res* 108, 917-928 (2011).
2. Durham AL, Speer MY, Scatena M, Giachelli CM, Shanahan CM. Role of smooth muscle cells in vascular calcification: implications in atherosclerosis and arterial stiffness. *Cardiovasc Res* 114, 590-600 (2018).
3. Iyemere VP, Proudfoot D, Weissberg PL, Shanahan CM. Vascular smooth muscle cell phenotypic plasticity and the regulation of vascular calcification. *J Intern Med* 260, 192-210 (2006).
4. Hortells L, Sur S, St Hilaire C. Cell Phenotype Transitions in Cardiovascular Calcification. *Front Cardiovasc Med* 5, 27 (2018).
5. Rank L, et al. Analyzing structure-function relationships of artificial and cancer-associated PARP1 variants by reconstituting TALEN-generated HeLa PARP1 knock-out cells. *Nucleic Acids Res* 44, 10386-10405 (2016).
6. Langelier MF, Planck JL, Roy S, Pascal JM. Structural basis for DNA damage-dependent poly(ADP-ribosylation) by human PARP-1. *Science* 336, 728-732 (2012).
7. Dawicki-McKenna JM, et al. PARP-1 Activation Requires Local Unfolding of an Autoinhibitory Domain. *Mol Cell* 60, 755-768 (2015).
8. Huang D, Yang CZ, Yao L, Wang Y, Liao YH, Huang K. Activation and overexpression of PARP-1 in circulating mononuclear cells promote TNF-alpha and IL-6 expression in patients with unstable angina. *Arch Med Res* 39, 775-784 (2008).

9. Martinez-Zamudio RI, Ha HC. PARP1 enhances inflammatory cytokine expression by alteration of promoter chromatin structure in microglia. *Brain Behav* 4, 552-565 (2014).
10. Bai P, Virag L. Role of poly(ADP-ribose) polymerases in the regulation of inflammatory processes. *FEBS Lett* 586, 3771-3777 (2012).
11. Hassa PO, Hottiger MO. A role of poly (ADP-ribose) polymerase in NF-kappaB transcriptional activation. *Biol Chem* 380, 953-959 (1999).
12. Zhang Y, et al. Inhibition of Poly(ADP-Ribose) Polymerase-1 Protects Chronic Alcoholic Liver Injury. *Am J Pathol* 186, 3117-3130 (2016).
13. Tanaka T, et al. Runx2 represses myocardin-mediated differentiation and facilitates osteogenic conversion of vascular smooth muscle cells. *Mol Cell Biol* 28, 1147-1160 (2008).
14. Jono S, et al. Phosphate regulation of vascular smooth muscle cell calcification. *Circ Res* 87, E10-17 (2000).
15. Vaziri ND. Oxidative stress in uremia: nature, mechanisms, and potential consequences. *Semin Nephrol* 24, 469-473 (2004).
16. Vaziri ND, Dicus M, Ho ND, Boroujerdi-Rad L, Sindhu RK. Oxidative stress and dysregulation of superoxide dismutase and NADPH oxidase in renal insufficiency. *Kidney Int* 63, 179-185 (2003).
17. Wu D, et al. Hydrogen sulfide ameliorates chronic renal failure in rats by inhibiting apoptosis and inflammation through ROS/MAPK and NF-kappaB signaling pathways. *Sci Rep* 7, 455 (2017).
18. Vavrinc P, van Dokkum RP, Goris M, Buikema H, Henning RH. Losartan protects mesenteric arteries from ROS-associated decrease in myogenic constriction following 5/6 nephrectomy. *J Renin Angiotensin Aldosterone Syst* 12, 184-194 (2011).
19. Luo X, Kraus WL. On PAR with PARP: cellular stress signaling through poly(ADP-ribose) and PARP-1. *Genes Dev* 26, 417-432 (2012).
20. Rodriguez-Vargas JM, et al. ROS-induced DNA damage and PARP-1 are required for optimal induction of starvation-induced autophagy. *Cell Res* 22, 1181-1198 (2012).

Reviewers' comments:

Reviewer #1 (Remarks to the Author):

The authors have undergone extensive efforts to add additional experiments and significantly improved the manuscript and addressed most noticed shortcomings.

However, the statistical section should be rechecked, maybe by a statistician, as scedasticity and normality are now confusing. Also, SEM was suddenly switched to SD, but the error bars remained the same?

The discussion on NF-kB is very speculative. It should at least be better supported by citations about the role and regulation of NF-kB in vascular calcification.

Reviewer #2 (Remarks to the Author):

The authors have revised and improved the manuscript extensively in most parts according to the comments from the reviewers. Following minor points should be also checked and revised.

-Fig. 5J: the photos of von Kossa and HE images of the third top and bottom panels from the right do not seem to match and should be checked.

-Legend of Supplementary Fig. 6: the description of the P value for asterisk should be added.

-Please correct the typos including:

1) sentence at line 669: "The binding ability of miR-204 to Runx2 3'UTR in RISC complex were detected"

2) sentence at line 672: "osteogenic genes (OPN, ColIA1, OC and Runx2) expression were"

Reviewer #3 (Remarks to the Author):

no further comments

Reviewer #1 (Remarks to the Author)

The authors have undergone extensive efforts to add additional experiments and significantly improved the manuscript and addressed most noticed shortcomings.

However, the statistical section should be rechecked, maybe by a statistician, as scedasticity and normality are now confusing. Also, SEM was suddenly switched to SD, but the error bars remained the same?

Response: Thanks for your suggestions. We have invited Dr. Yuelin Chao and Dr. Ramprasath Tharmarajan to help us recheck the statistical analysis.

Firstly, the homogeneity of variance was assessed by the F test (two groups) or Brown-Forsythe test (≥ 3 groups). Then, for parametric data, Student's t test was used to analyze the statistical significances of differences between two groups, and ANOVA to analyze the statistical significances of differences among multiple groups. For non-parametric data, Mann-Whitney U test was used to analyze the statistical significances of differences between two groups, and Kruskal-Wallis test followed by the Dunn's post hoc test was used among multiple groups.

Actually, values of all the data in this research were shown as the means \pm S.D. of at least three independent experiments. We wrongly labelled SD as SEM in the version of initial peer review. In the first revision, we have checked and corrected the mistake. We feel sorry for the carelessness.

The discussion on NF- κ B is very speculative. It should at least be better supported by citations about the role and regulation of NF- κ B in vascular calcification.

Response: Yes, We have added more discussion about the role and regulation of NF- κ B in vascular calcification.

NF- κ B has been implicated in promoting high phosphate- induced VSMC calcification^{1, 2, 3}. Inhibition of NF- κ B activity within SMCs reduces arterial medial calcification in mice with chronic kidney disease⁴. We and other researchers have revealed NF- κ B as a substrate of PARP1-mediated poly(ADP-ribosyl)ation, which then enhanced NF- κ B-dependent promoter^{5, 6}. And IL-6 is a classical direct target for activated NF- κ B^{7, 8, 9}. Interestingly, in this research, we found that PARP1 deficiency decreased, but PARP1 overexpression increased, both the mRNA and protein levels of IL-6 in arteries of CRF rats *in vivo*, all supposing that PARP1 possibly upregulates IL-6 expression via promoting NF- κ B transactivation in VSMC calcification.

Reviewer #2 (Remarks to the Author)

The authors have revised and improved the manuscript extensively in most parts according to the comments from the reviewers. Following minor points should be also checked and revised.

-Fig. 5J: the photos of von Kossa and HE images of the third top and bottom panels from the right do not seem to match and should be checked.

-Legend of Supplementary Fig. 6: the description of the P value for asterisk should be added.

-Please correct the typos including:

1) sentence at line 669: "The binding ability of miR-204 to Runx2 3'UTR in RISC complex were detected"

2) sentence at line 672: "osteogenic genes (OPN, ColIA1, OC and Runx2) expression were"

Response: We have revised the points accordingly. Thank you for your good comments.

-The unmatched figure in Fig. 5J has been replaced.

-The description of the P value was added.

-The typos are corrected as following.

1) The binding ability of miR-204 to Runx2 3'UTR in RISC complex was detected

2) The expression of osteogenic genes (OPN, ColIA1, OC and Runx2) was analyzed by western blot.

Reference

1. Zhao G, *et al.* Activation of nuclear factor-kappa B accelerates vascular calcification by inhibiting ankylosis protein homolog expression. *Kidney Int* **82**, 34-44 (2012).
2. Al-Aly Z. Phosphate, oxidative stress, and nuclear factor-kappaB activation in vascular calcification. *Kidney Int* **79**, 1044-1047 (2011).
3. Zhao MM, *et al.* Mitochondrial reactive oxygen species promote p65 nuclear translocation mediating high-phosphate-induced vascular calcification in vitro and in vivo. *Kidney Int* **79**, 1071-1079 (2011).
4. Yoshida T, Yamashita M, Horimai C, Hayashi M. Smooth Muscle-Selective Nuclear Factor-kappaB Inhibition Reduces Phosphate-Induced Arterial Medial Calcification in Mice With Chronic Kidney Disease. *J Am Heart Assoc* **6**, (2017).
5. Hassa PO, Hottiger MO. A role of poly (ADP-ribose) polymerase in NF-kappaB transcriptional activation. *Biol Chem* **380**, 953-959 (1999).
6. Zhang Y, *et al.* Inhibition of Poly(ADP-Ribose) Polymerase-1 Protects Chronic Alcoholic Liver Injury. *Am J Pathol* **186**, 3117-3130 (2016).
7. Huang D, Yang CZ, Yao L, Wang Y, Liao YH, Huang K. Activation and overexpression of PARP-1 in circulating mononuclear cells promote TNF-alpha and IL-6 expression in patients with unstable angina. *Arch Med Res* **39**, 775-784 (2008).
8. Martinez-Zamudio RI, Ha HC. PARP1 enhances inflammatory cytokine expression by alteration of promoter chromatin structure in microglia. *Brain Behav* **4**, 552-565 (2014).
9. Bai P, Virag L. Role of poly(ADP-ribose) polymerases in the regulation of inflammatory processes. *FEBS Lett* **586**, 3771-3777 (2012).

REVIEWERS' COMMENTS:

Reviewer #1 (Remarks to the Author):

The authors extended their discussion on NF- κ B. The statistical section apparently has been rechecked, yet remains a bit cryptic to me.

Reviewer #1 (Remarks to the Author):

The authors extended their discussion on NF-kB. The statistical section apparently has been rechecked, yet remains a bit cryptic to me.

Response: We are very sorry for statistical section that has been not explained clear for you.

We have revised the manuscript as follow:

Statistical analysis:

Values are shown as the means \pm S.D. of at least three independent experiments. Normality of data distribution was assessed by the Shapiro–Wilk test prior to the application of parametric tests. For non-normally distributed data, nonparametric tests were used to analyze statistical differences. For comparisons between two groups, significance was determined using Student’s t-test or nonparametric Mann–Whitney test. For comparisons among multiple groups, ANOVA followed by posthoc Bonferroni test or nonparametric Kruskal-Wallis test followed by the Dunn’s post hoc test. An F test (two groups) or Brown-Forsythe test (multiple groups) was used to determine difference in variances for t-test and ANOVA, respectively. The statistical significance of correlations was determined by Pearson’s correlation coefficient analysis. Significant differences are indicated by * or # ($P < 0.05$), and very significant differences are indicated by ** or ## ($P < 0.01$). All statistical analyses were performed using SPSS software (version 22.0, SPSS Inc).